# Cytokinin depends on GA biosynthesis and signaling to regulate different aspects of vegetative phase change in Arabidopsis

Sören Werner [1], Danuše Tarkowskà [2] & Thomas Schmülling [1] ✉

The vegetative juvenile-to-adult transition (vegetative phase change) is a critical phase in plant development, the timing of which is controlled by the highly conserved age pathway, comprising the miR156/miR157-SPL module and the downstream miR172-AP2-like module, and is modulated by exogenous and endogenous cues. The phytohormones cytokinin (CK) and gibberellin (GA) have been described to both alter miR172 levels, most probably by regulating SPL activity. In this study, we establish an epistatic relation between CK and GA, in which CK action depends on GA, contrasting with the antagonistic nature described previously for CK-GA crosstalk. We show that CK positively affects GA biosynthesis during *Arabidopsis* vegetative development and depends on the GA biosynthetic enzymes GA3ox1 and GA3ox2 to modify the appearance of abaxial trichomes as well as leaf shape, both hallmarks of vegetative phase change. Downstream of CK, epidermal identity is regulated in dependence of SPL transcription factors, the GA signaling repressors GAI and RGA and the miR172-targets TOE1 and TOE2. Notably, genetic analysis revealed that GA regulates this process also CK-independently. Furthermore, our data from genetic analyses suggests that CK affects leaf shape through other GA signaling components and AP2-like transcription factors rather than SPLs. Hence, CK differentially regulates several aspects of vegetative phase change. The work contributes to the understanding of vegetative phase change regulation as well as phytohormone crosstalk in general.

Plants progress through several developmental phases during their life cycle. Following germination, the shoots of most plants pass through a vegetative phase before undergoing a transition to reproductive growth. During vegetative growth, plants increase their photosynthetic capacity and their biomass to meet the energy-sapping demands of reproduction. The correct timing of the transitions from one phase to the next is essential to ensure proper development and reproductive success[1,2].

Vegetative development can be further divided into a juvenile and an adult phase. The juvenile-to-adult transition (vegetative phase change, VPC) and the accompanying heteroblastic features, as well as the acquisition of reproductive competence, are regulated by the age-dependent pathway, consisting of the microRNAs miR156/miR157 and miR172 and their respective target genes[1,3]. An initially high abundance of miR156 and miR157 ensures juvenility in the seedling stage[4,5]. The transition to the adult vegetative stage is achieved by a miR156/miR157 decline with advancing age, allowing an increase in the expression of their target genes, which belong to the *SQUAMOSA promoter binding protein-like (SPL)* family[5–8]. MiR156/miR157 target the transcripts of ten out of 16 *SPL* transcription factor genes, six of which promote leaf

[1]Institute of Biology/Applied Genetics, Dahlem Centre of Plant Sciences (DCPS), Freie Universität Berlin, Albrecht-Thaer-Weg 6, Berlin, Germany. [2]Laboratory of Growth Regulators, Institute of Experimental Botany, Czech Academy of Sciences and Faculty of Sciences, Palacký University, Šlechtitelů 27, Olomouc, Czech Republic. ✉e-mail: t.schmuelling@fu-berlin.de

traits characteristic of the adult vegetative phase[9]. The increase in SPL activity causes in *Arabidopsis* successive rosette leaves to become bigger and more elongated in shape, the serration of the margins increases, cell size decreases, and trichomes appear on the lower surface[10–13].

At least five of miR156/miR157-targeted SPLs are directly involved in promoting the transcription of *MIR172* genes[9]. Hence, miR172 abundance increases as the shoot develops, which in turn gradually inhibits the accumulation of the targets APETALA2 (AP2) and the AP2-LIKE proteins SCHLAFMÜTZE, SCHNARCHZAPFEN, TARGET OF EAT1 (TOE1), TOE2, and TOE3[14–17]. Despite the more or less linear nature of the age pathway[5,18], a certain independence of the two modules in regulating VPC can be seen in the fact that SPLs promote most, if not all leaf traits characteristic for the adult vegetative phase, whereas the miR172-AP2-like module affects epidermal identity rather than leaf morphology[5,9,12,18,19]. Furthermore, several studies indicate that the miR172-AP2-like module is also regulated independently of miR156 by other factors[3,20–24].

The timing of the juvenile-to-adult transition is characterized by the heteroblastic features regulated by the age pathway. The appearance of trichomes on the lower side of the leaf blade is controlled by SPLs, in part through their effect on miR172 expression[3,5,6,25], as well as miR172-targeted *AP2-like* genes. Except for *TOE3*, there is clear evidence for the involvement of all miR172 targets in abaxial trichome production, with *TOE1* and *TOE2* having the strongest impact[18,20,26]. Their interaction with the leaf polarity factor KAN1 inhibits abaxial expression of *GLABRA1*, a gene well known for its role in trichome formation. The age-dependently increasing repression of AP2-like factors by miR172 determines the timing of adult epidermal identity[26–28]. An often-used additional parameter for VPC is the age-dependent change of leaf shape, characterized by an increase in the length-to-width ratio of the leaf blade[8,29,30]. It is achieved by the SPL9/SPL13-mediated downregulation of *BOP1/BOP2* in successive leaves, resulting in delayed establishment of proliferative regions in leaves, which promotes expansion of the leaf blade[31–33].

Among the five *MIR172* genes in *Arabidopsis*, *MIR172A* and *MIR172B* play dominant roles in the timing of trichome initiation[34]. Both genes were shown to be induced by the phytohormone cytokinin (CK), which positively regulates VPC involving SPLs as well as the miR172 targets *TOE1* and *TOE2*[20]. CKs are a group of $N^6$-substituted adenine derivatives that are perceived in *Arabidopsis* by three membrane-bound histidine kinase receptors, namely CYTOKININ RESPONSE 1 (CRE1), ARABIDOPSIS HISTIDINE KINASE2 (AHK2), and AHK3[35,36], with the latter two playing a more crucial role in the CK-dependent regulation of VPC[20]. The CK signal is transduced via a multi-step His-Asp phosphorelay by a two-component signaling system, eventually activating type-B response regulators (ARRs) that act as transcription factors to mediate the CK response[37]. Among the 11 type-B ARRs found in *Arabidopsis*, ARR1, ARR10 and ARR12 were identified to take part in phase change regulation[20].

Also gibberellins (GAs) have been found to regulate VPC in several species[38–40]. GA is essential for the juvenile-to-adult phase transition in *Arabidopsis* since under short-day (SD) conditions, *ga1* mutants, which contain nearly no GA, do not produce abaxial trichomes at all and are unable to achieve reproductive competence[41–43]. GAs are a class of diterpenoid plant hormones and are perceived by three nuclear GA INSENSITIVE DWARF1 (GID1) receptor isoforms (GID1A, GID1B, GID1C). Upon GA binding, GID1 undergoes conformational changes that facilitate the interaction with DELLA proteins, which are subsequently degraded[44–48]. The *Arabidopsis* genome encodes five DELLA proteins, GIBBERELLIC ACID INSENSITIVE (GAI), REPRESSOR OF GA1-3 (RGA), RGA-LIKE1 (RGL1), RGL2 and RGL3, which display partially redundant functions in modulating vegetative and reproductive growth[41,49–53]. Among them, GAI and RGA have been shown to be involved in abaxial trichome formation[10,41,42].

Similar to CK, GA positively affects *MIR172B* expression and miR172 levels[54,55]. Furthermore, both CK and GA do not affect miR156 abundance[6,20,55] and 35S:MIR156 plants contain the same concentrations of CK and GA as the wild type[54,56], indicating that there is no close regulatory link between miR156 and these two hormones. Transcript levels of *SPL* genes involved in VPC regulation (*SPL2, SPL9, SPL10, SPL11, SPL13, SPL15*) are largely unaffected in seedlings by either CK or GA treatment, or by a reduction of endogenous hormone levels or signaling[6,9,20,55]. But SPL participation in the CK- and GA-dependent regulation of VPC is suggested to take place on the post-translational level: On the one hand, no reduction in juvenile leaf number by an increased CK status is observed in the background of 35S:MIR156, which lacks SPL activity. On the other hand, the inhibition of *MIR172B* transcriptional activation by RGA is abrogated by its interaction with SPL9[20,54]. Also type-B ARRs were shown to interact with SPLs from the above-mentioned group[56], but the significance of this observation for the regulation of VPC remains elusive so far.

The apparent similarities of how CK and GA pathways act on the age pathway and regulate the juvenile-to-adult phase transition led us to investigate a possible crosstalk between the two hormones in this process. Using genetic interaction studies and expression analyses, we discovered that CK depends on GA biosynthesis and signaling to exert its influence on VPC, with particular importance of the GA biosynthesis genes *GA3ox1* and *GA3ox2*, and the DELLA proteins GAI and RGA. Our work establishes an epistatic relationship between CK and GA, in which both hormones have a positive influence on VPC, but CK signaling is hypostatic to GA signaling. This contrasts with the hitherto existing view that interaction between the two hormones is antagonistic[57–59].

## Results

### GA partially compensates for a low CK status in the regulation of VPC

Both CK and GA are positive regulators of VPC[20,41,60]. To test the interdependence of the two hormones in regulating VPC, we crossed the *rock2* mutant expressing a constitutively active variant of the AHK2 receptor[61] with the GA biosynthesis mutant *ga1*, which lacks the *ent*-copalyl diphosphate synthase (CPS) enzyme for the first and rate-limiting step in GA biosynthesis[62]. The *rock2* mutant produced less leaves without abaxial trichomes compared to the wild type ($6.2 \pm 0.2$ compared to $7.8 \pm 0.2$), whereas *ga1* failed to produce abaxial trichomes at all. *Rock2* was not able to restore the ability of *ga1* to do so (Fig. 1a).

In our previous study, we focused on epidermal identity to define the timing of VPC[20]. However, because of SPL involvement in VPC regulation by CK and GA, we also suspected that other morphological features controlled by SPLs besides abaxial trichome appearance would be affected. To investigate the extent to which CK regulates other aspects of VPC, we analyzed the ratio of length to width of the leaf blade as an additional marker of VPC. Interestingly, the leaf length-width ratio of *ga1* was not statistically different from the wild type. The *rock2* mutant had longer leaves in general, but did not have this effect in the *ga1* background (Fig. 1b). This result complements those shown in Fig. 1a.

We also introgressed the *rock2* mutation into *ga3ox1 ga3ox2*, which has a reduced level of bioactive GAs, especially of $GA_4$[63], and showed a strongly retarded VPC (Fig. 1c, d). The *rock2* mutant was neither able to reduce the number of leaves not forming abaxial trichomes in *ga3ox1 ga3ox2* (Fig. 1c) nor to compensate for the overall rounder leaf shape of *ga3ox1 ga3ox2* (Fig. 1d).

These results suggest that GA acts downstream of CK signaling. If that is the case, exogenously applied GA should be able to rescue the phenotype of plants that are CK-deficient due to lower CK levels or signaling. To test this, the receptor mutant *ahk2 ahk3*, the type-B *ARR* mutant *arr1,10,12* and plants overexpressing a gene encoding a CK degrading enzyme (CKX1ox) were treated with solutions of bioactive

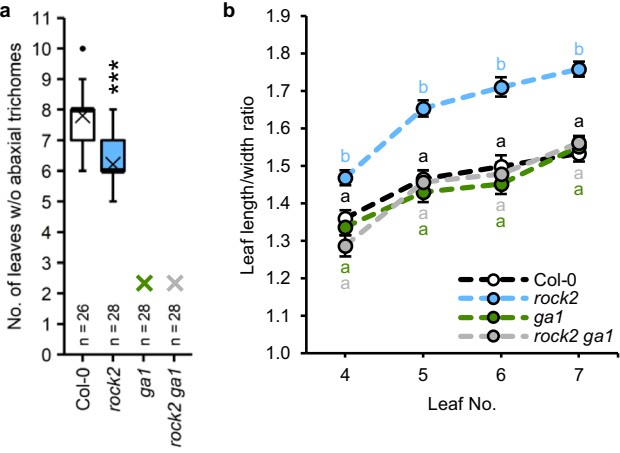

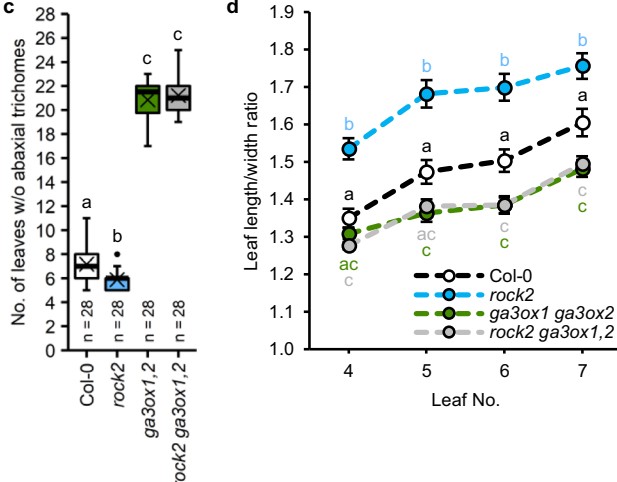

**Fig. 1 | The vegetative phase change phenotype of GA biosynthesis mutants is not rescued by enhanced CK signaling. a, c** Number of leaves without abaxial trichomes of SD-grown *rock2 ga1* (**a**) and *rock2 ga3ox1,2* (**c**) hybrid plants in comparison to their respective parents and wild type. In box plots, the center line represents the median value and the boundaries indicate the 25th percentile (upper) and the 75th percentile (lower). The X marks the mean value. Whiskers extend to the largest and smallest value, excluding outliers which are shown as dots. **b, d** Length-to-width ratios of the blades of leaves 4 to 7. Data displayed are expressed as mean ± SEM of SD-grown plants. Numbers of biological replicates: (**b**)

Col-0 ($n_4 = 32$; $n_5 = 32$; $n_6 = 33$; $n_7 = 32$), *rock2* ($n_4 = 32$; $n_5 = 31$; $n_6 = 30$; $n_7 = 32$), *ga1* ($n_4 = 38$; $n_5 = 38$; $n_6 = 38$; $n_7 = 38$), *rock2 ga1* ($n_4 = 29$; $n_5 = 29$; $n_6 = 30$; $n_7 = 30$); (**d**) Col-0 ($n_4 = 28$; $n_5 = 28$; $n_6 = 29$; $n_7 = 29$), *rock2* ($n_4 = 24$; $n_5 = 26$; $n_6 = 27$; $n_7 = 27$), *ga3ox1 ga3ox2* ($n_4 = 30$; $n_5 = 32$; $n_6 = 32$; $n_7 = 32$), *rock2 ga3ox1,2* ($n_4 = 32$; $n_5 = 32$; $n_6 = 31$; $n_7 = 31$). Asterisks indicate statistically significant differences compared to the wild type, as calculated by Mann-Whitney test (***$p < 0.001$) (**a**); letters indicate statistically significant differences between the genotypes, as calculated by one-way ANOVA, post-hoc Tukey's test ($p < 0.05$) (**b, d**) or Kruskal-Wallis test ($q < 0.05$) (**c**).

GAs, the $C_{13}$-hydroxylated $GA_3$ or the non-$C_{13}$-hydroxylated $GA_{4+7}$ (Fig. 2).

Interestingly, in contrast to the experiment shown in Fig. 1a, mock-treated *ga1* was capable of progressing into the adult phase, indicating that the treatment itself circumvents the necessity of GA for transitioning. Nevertheless, mock-treated *ga1* produced five times more juvenile leaves than mock-treated wild-type plants (48.4 ± 1.5 compared to 9.4 ± 0.4). All CK-deficient lines showed a delay in the appearance of abaxial trichomes as previously described[20]. Treatment with either $GA_3$ or $GA_{4+7}$ caused a reduction in juvenile leaf number of the wild type and completely rescued the phenotype of all tested mutant lines (Fig. 2a), supporting the idea of CK depending on GA in the regulation of VPC.

Similar to epidermal identity, mock-treated *ga1* also behaved differently in terms of leaf shape and produced rounder leaves compared to the wild type (Fig. 2b). This was fully compensated by exogenous GA treatment. The CK-deficient genotypes showed all rounder leaves than the wild type, suggesting a more juvenile status. Among these genotypes, GA treatment fully restored the leaf shape of *arr1,10,12* to the level of mock-treated wild type, whereas only partial rescue was observed for *ahk2 ahk3* and CKX1ox (Fig. 2b).

## CK promotes GA biosynthesis

If CK acts upstream of GA, it could act through influencing GA metabolism and/or signaling. First, we tested the impact of a reduced CK content or signaling on the expression of GA biosynthesis genes in shoots of young plants (Fig. 3a, b, d, e, g, h) as well as the inducibility of these genes by exogenous application of CK (Fig. 3c, f, i).

The initial and rate-limiting steps of GA biosynthesis are carried out by two terpene synthases (CPS = GA1, and *ent*-kaurene synthase, KS = GA2)[64], followed by two types of cytochrome P450 monooxygenases (*ent*-kaurene oxidase, KO = GA3, and *ent*-kaurenoic acid oxidase (KAO)). The sequential action of these enzymes converts geranylgeranyl diphosphate to $GA_{12}$, which is the precursor for all other GA metabolites[65]. In shoots of 7-, 14-, and 21-day-old SD-grown plants with a lower CK content (CKX1ox), we observed a downregulation of the genes coding for the GA1 and GA2 enzymes. Both

genes were also induced by 6-benzyladenine (BA) treatment of wild-type seedlings (Fig. 3a–c). In contrast, CK deficiency (CKX1ox, *ahk2 ahk3*) or treatment did not affect the transcript levels of *GA3*, *KAO1* and *KAO2* (Fig. 3c, Supplementary Fig. 1).

Bioactive GA metabolites are produced from $GA_{12}$ by the concerted action of GA20ox and GA3ox enzymes[65]. *GA20ox1* was downregulated in the CK-deficient lines at all time points, *GA20ox2* at least at the latest time point 21 days after germination (DAG) (Fig. 3d, e) and both genes were rapidly and strongly induced by CK (Fig. 3f). *GA3ox1* and *GA3ox2* also showed reduced expression in plants with a lower CK status (Fig. 3g, h) and an upregulation after CK treatment (Fig. 3i).

In conclusion, CK promotes the expression of genes responsible for the first rate-limiting steps as well as the production of the bioactive forms during vegetative growth. The functional relevance of this influence for regulating VPC is supported by the observation that enhanced CK signaling has no impact on juvenile leaf number or leaf shape in the background of *ga1* and *ga3ox1 ga3ox2* (Fig. 1).

Bioactive GAs are deactivated through 2β-hydroxylation by gibberellin 2-oxidases (GA2ox)[65,66]. Interestingly, *GA2ox1* was also downregulated in *ahk2 ahk3* and CKX1ox. Furthermore, *GA2ox2* also showed reduced expression in CK-deficient genotypes compared to the wild type, but only at the earliest time point 7 DAG (Supplementary Fig. 2a, b). Whether the reduced *GA2ox* expression reflects a direct CK effect or a mechanism responding to increased GA biosynthesis remains to be shown. The rapid increase of *GA2ox1* transcript levels in response to CK treatment (Supplementary Fig. 2c) indicates a rather direct effect.

In order to assess the consequence of the CK-dependent transcript changes for the GA content in the plant shoot, we determined the concentrations of GA biosynthetic precursors ($GA_{9,12,13,15,19,20,24,44,53}$), bioactive GAs ($GA_{1,3,4,5,6,7}$) and deactivated GAs ($GA_{8,29,34,51}$) in the same type of material as used for the gene expression analysis (Supplementary Table 1). The concentrations of total GA or bioactive forms of GA were not changed at none of the time points (Supplementary Fig. 3a, b).

$GA_{12}$ is the precursor of all GA metabolites not hydroxylated at $C_{13}$, including the bioactive forms $GA_4$ and $GA_7$. Furthermore, hydroxylation of $GA_{12}$ results in $GA_{53}$, which is the precursor of all $C_{13}$-

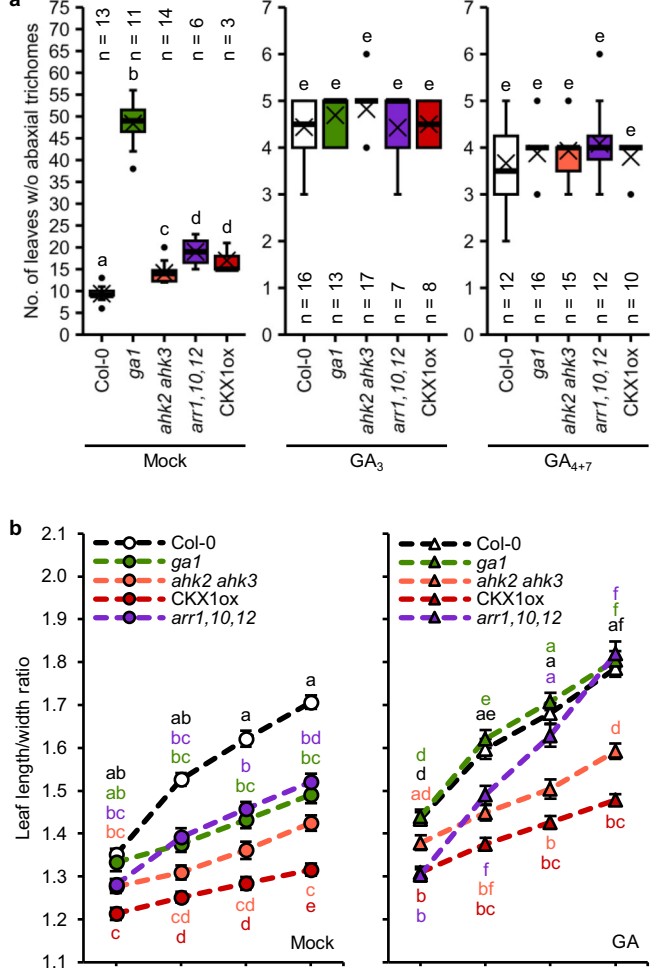

**Fig. 2 | Exogenously applied GA partially rescues the delayed vegetative phase change of CK-deficient plants. a** Number of leaves without abaxial trichomes of SD-grown wild-type and cytokinin mutant plants treated with GA compared to the mock-treated control. In box plots, the center line represents the median value and the boundaries indicate the 25th percentile (upper) and the 75th percentile (lower). The X marks the mean value. Whiskers extend to the largest and smallest value, excluding outliers which are shown as dots. **b** Length-to-width ratios of the blades of leaves 4 to 7 of mock- and $GA_{4+7}$-treated plants. Data displayed are expressed as mean ± SEM of SD-grown plants. Numbers of biological replicates: Col-0 (Mock: $n_4 = 32$; $n_5 = 33$; $n_6 = 33$; $n_7 = 33$; GA: $n_4 = 37$; $n_5 = 36$; $n_6 = 37$; $n_7 = 36$), ga1 (Mock: $n_4 = 37$; $n_5 = 36$; $n_6 = 37$; $n_7 = 37$; GA: $n_4 = 35$; $n_5 = 34$; $n_6 = 36$; $n_7 = 34$), ahk2 ahk3 (Mock: $n_4 = 34$; $n_5 = 34$; $n_6 = 35$; $n_7 = 34$; GA: $n_4 = 41$; $n_5 = 42$; $n_6 = 41$; $n_7 = 42$), arr1,10,12 (Mock: $n_4 = 32$; $n_5 = 29$; $n_6 = 32$; $n_7 = 31$; GA: $n_4 = 36$; $n_5 = 37$; $n_6 = 37$; $n_7 = 37$), CKX1ox (Mock: $n_4 = 30$; $n_5 = 29$; $n_6 = 30$; $n_7 = 30$; GA: $n_4 = 53$; $n_5 = 50$; $n_6 = 53$; $n_7 = 53$). Letters indicate statistically significant differences regarding genotypes and treatments, as calculated by two-way ANOVA, post-hoc Tukey's test ($p < 0.05$) (**a**, **b**).

hydroxylated GA metabolites, including the bioactive forms $GA_1$, $GA_3$, $GA_5$ and $GA_6$[65]. The sum of 13-hydroxylated forms was slightly elevated in CK-deficient lines at 7 DAG compared to the wild type, while the concentration of 13-non-hydroxylated forms was considerably reduced at that time point (Supplementary Fig. 3c, d). To investigate the latter observation further, we had a look at the non-13-hydroxylated metabolites in more detail (Fig. 4). The concentrations of $GA_{12}$, $GA_{15}$ and $GA_7$ were below the detection limit and $GA_{24}$ and $GA_9$ could not be determined in every biological replicate (Supplementary Table 1). No significant difference between the genotypes was detected

for $GA_9$, the direct precursor of bioactive $GA_4$. Deactivation of $GA_9$ results in the production of $GA_{51}$ which was strongly increased in CK-deficient plants at the latest time point. $GA_4$ and in particular its degradation product $GA_{34}$, on the other hand, showed reduced levels compared to the wild type.

The reduced concentration of bioactive $GA_4$ is in agreement with the decreased expression of *GA3ox1* and *GA3ox2* in CK-deficient lines (Fig. 3g, h) and the requirement of both genes for CK-dependent regulation of VPC (Fig. 1c, d). Both enzymes were shown to be responsible for the production of most of $GA_4$ in *Arabidopsis* plants[63]. In summary, CK affects the concentrations of only a few metabolites, not changing the overall GA composition in the plant, indicating that the CK influence on GA biosynthesis is specific.

**The DELLA genes *GAI* and *RGA* are necessary and sufficient to mediate the influence of CK on epidermal identity, but not on leaf shape**

Next, we investigated on which GA signaling elements the promotion of VPC by CK depends. We crossed *rock2* with the GA triple receptor mutant *gid1a,b,c* in order to obtain combinations of *rock2* with the double and triple receptor mutants. However, among the segregating F2 progeny we could only identify *rock2 gid1b,c*. The *gid1b gid1c* mutant produces more rosette leaves until bolting[55], which might in part be due to an increase in juvenile leaf number (Supplementary Fig. 4). The *rock2 gid1b,c* mutant plants displayed a reduced number of juvenile leaves compared to *gid1b gid1c*, which indicates functional redundancy among the GA receptors in the CK-dependent regulation of VPC. In this case, the sole presence of GID1A appears to be sufficient to transmit the CK signal to downstream effectors. Alternatively, CK could also act GID1-independently.

Next, we crossed all five DELLA single mutants with *ahk2 ahk3*. The *rga* mutant was the only *della* mutant showing a reduced number of juvenile leaves ($6.3 \pm 0.1$ compared to $5.6 \pm 0.1$ in wild type) (Supplementary Fig. 5a). All other single *della* mutants (*gai, rgl1, rgl2, rgl3*) had a similar number of leaves without abaxial trichomes as wild type (Supplementary Fig. 5a, b). The *rga* mutation caused also a significantly earlier transition to the adult phase in the *ahk2 ahk3* background (Supplementary Fig. 5a). Although the *gai* mutant did not have a phenotype in the wild-type background, it reduced juvenile leaf number in *ahk2 ahk3* (Supplementary Fig. 5b). This suggests that suppression of GA signaling by GAI contributes to the late juvenile-to-adult transition under CK deficiency. In contrast, introgression of *rgl1, rgl2* or *rgl3* in the *ahk2 ahk3* background did not change the late appearance of abaxial trichomes in this mutant (Supplementary Fig. 5c).

We then generated *gai rga* and *ahk2,3 gai rga* lines and found that the delayed transition of *ahk2 ahk3* was completely lost when combined with *gai rga* (Fig. 5a). This result shows that regulation of abaxial trichome appearance by CK depends on GA signaling, acting through the DELLA proteins GAI and RGA. Interestingly, *gai rga* did not exhibit a leaf shape phenotype, neither in wild-type nor in CK-deficient background (Fig. 5b), suggesting that CK activity is, in this case, independent of GA signaling or masked by a higher redundancy among *DELLA* genes in leaf shape regulation.

**The age-dependent control of leaf shape is differently regulated by CK than epidermal identity**

In our previous study, we used *rock2* MIR156ox plants to demonstrate the requirement of SPL activity for the CK-dependent regulation of epidermal identity since knock-out mutants were not available for all *SPL* genes involved in VPC control at that time[20]. In *rock2* MIR156ox enhanced CK signaling did not rescue the epidermal identity phenotype[20], because miR156 overexpression causes an extensive reduction of SPL activity[7,67,68]. In order to analyze in more detail the requirement of SPLs for VPC regulation by CK, we now introgressed

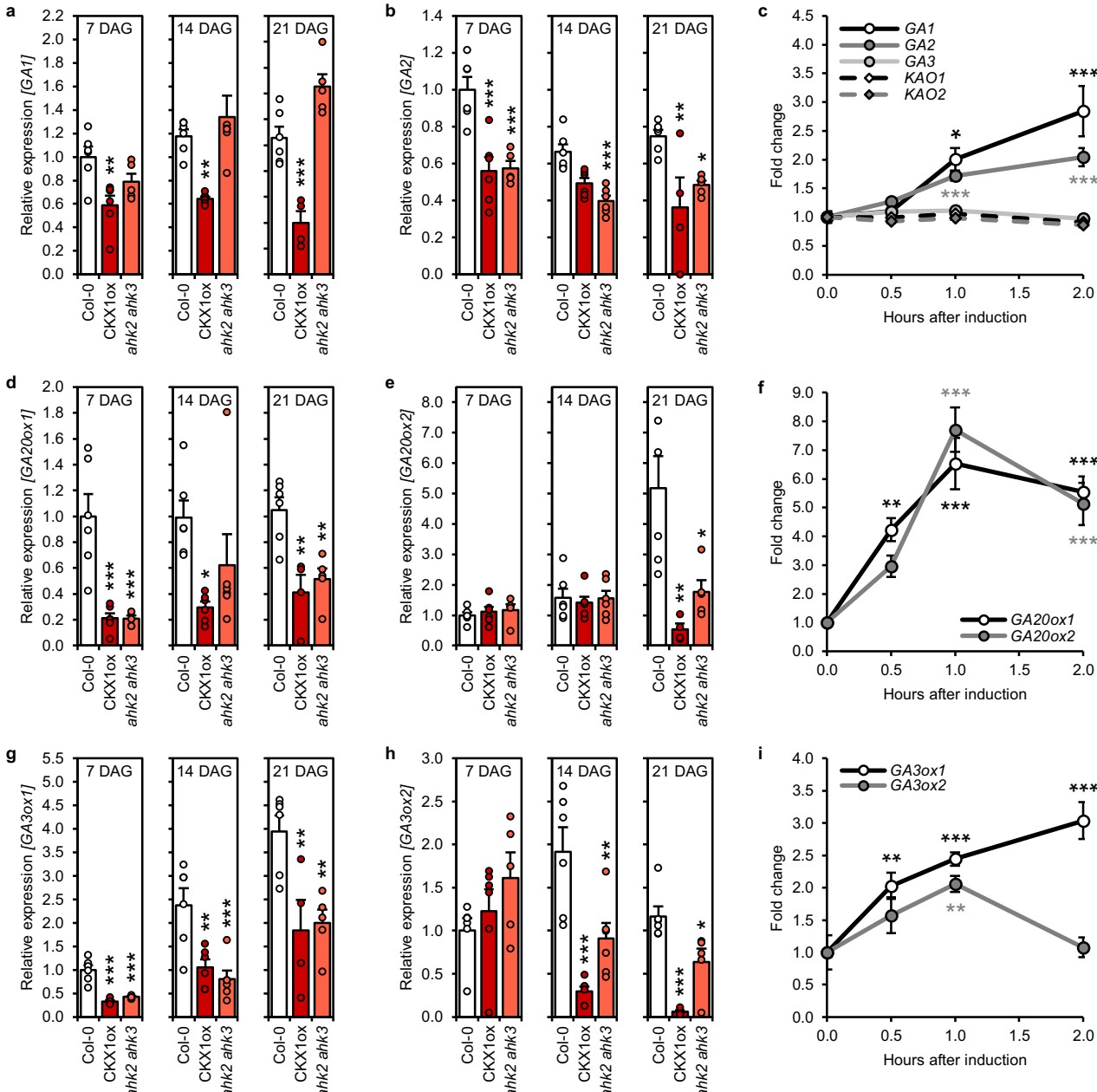

**Fig. 3 | Expression of GA biosynthesis genes is downregulated in CK-deficient plants. a, b, d, e, g, h** Expression of GA biosynthesis genes in whole shoots of SD-grown *ahk2 ahk3* and CKX1ox plants compared to the wild type. Numbers of biological replicates: Col-0 ($n_7$ = 6; $n_{14}$ = 6; $n_{21}$ = 6), CKX1ox ($n_7$ = 6; $n_{14}$ = 6; $n_{21}$ = 4), *ahk2 ahk3* ($n_7$ = 5; $n_{14}$ = 6; $n_{21}$ = 5). Dots indicate each single biological replicate. **c, f, i** Expression kinetics of GA biosynthesis genes in 10-day-old SD-grown wild-type seedlings after treatment with 1 μM BA ($n$ = 6 biological replicates). Transcript levels were determined by qRT-PCR. Data were normalized to *TAFII15* and *PP2AA2*. Data displayed are expressed as mean ± SEM. Asterisks indicate statistically significant differences compared to the wild type of the respective time point (**a, b, d, e, g, h**) or compared to time point 0 (**c, f, i**), as calculated by one-way ANOVA, post-hoc Dunnett's test (*$p < 0.05$; **$p < 0.01$; ***$p < 0.001$).

the *rock2* mutation into *spl2,9,10,11,13,15* as well as *spl2,10,11,13,15* retaining a functional SPL9. In the background of the quintuple *SPL* mutant, *rock2* was still able to reduce juvenile leaf number and to influence leaf shape, although to a lesser extent than in wild type (Supplementary Fig. 6a, b). The additional knock-out of *SPL9* substantially increased the number of leaves without abaxial trichomes compared to the quintuple mutant and *rock2* lost its effect on this trait (Supplementary Fig. 6c). This outcome confirmed the result obtained for *rock2* MIR156ox[20], and identified SPL9 as a downstream target of CK in epidermal identity regulation. Surprisingly, enhanced CK signaling only partially complemented the more roundish leaf shape phenotype

of the *spl* sixtuple mutant (Supplementary Fig. 6d), indicating that CK is not solely dependent on the tested *SPL* genes in this process.

We previously showed that CK depends on the miR172 targets TOE1 and TOE2 to regulate VPC[20]. The delayed appearance of abaxial trichomes in *ahk2 ahk3* plants is completely suppressed by the early-transitioning phenotype of *toe1 toe2* (Fig. 5a). *Toe1 toe2* did not show a leaf shape phenotype in the wild-type background indicating that regulation of leaf shape during VPC is different from that of epidermal identity. However, in the CK-deficient background of *ahk2 ahk3* lack of TOE1/TOE2 function increased the leaf length-to-width ratio, revealing a latent function of these transcriptional regulators and challenging

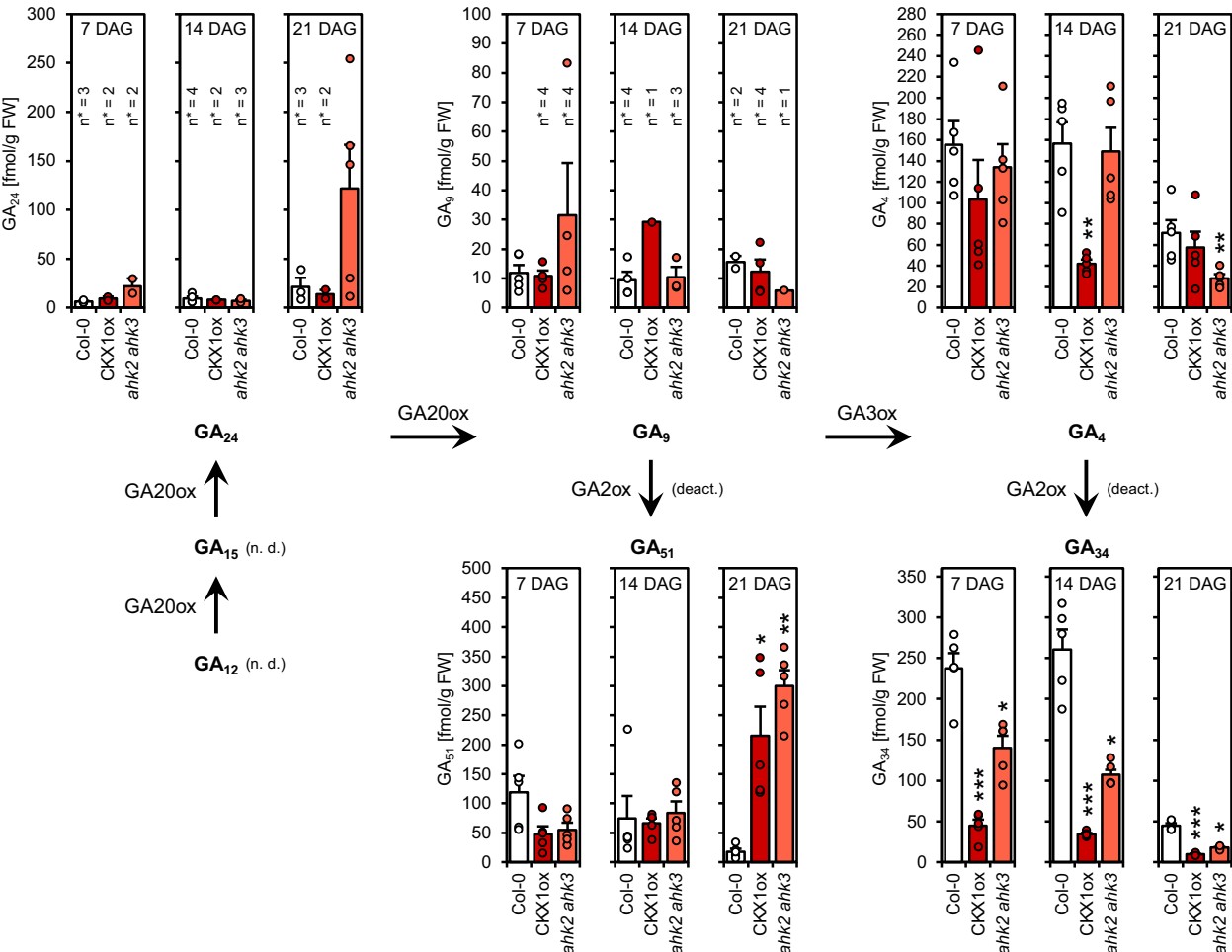

**Fig. 4 | Concentration of non-13-hydroxylated GA metabolites in CK-deficient plants.** Shown are concentrations of non-13-hydroxylated GA metabolites in whole shoots of SD-grown plants ($n = 5$ biological replicates). Data displayed are expressed as mean ± SEM. Dots indicate each single biological replicate. Asterisks indicate statistically significant differences compared to the wild type of the respective time point, as calculated by Kruskal-Wallis test (*$q < 0.05$; **$q < 0.01$; ***$q < 0.001$). n. d. = not detected; deact. = deactivation; n* = number of samples in which the respective GA metabolite could be detected in.

the notion that the age-dependent control of leaf morphology is solely mediated by the miR156-SPL module[5,9,12,18,19].

To address the question if GA depends on the same miR172 targets as CK, we crossed *gai rga* with *toe1 toe2*. Interestingly, the quadruple mutant displayed a slight but statistically significant further decrease in juvenile leaf number compared to *toe1 toe2* plants (Fig. 5a), indicating that additional *DELLA* and/or *AP2-like* genes participate in VPC control by GA. Introgression of *ahk2 ahk3* did not change the number of juvenile leaves of *gai rga toe1,2*, confirming that only these four genes are required and sufficient for the CK-dependent regulation of epidermal identity. The leaf shape of *gai rga toe1,2* was similar to the parental lines and wild-type plants. The leaves of *ahk2,3 gai rga toe1,2* resembled the ones of *ahk2,3 toe1,2* (Fig. 5b). This result confirms the role of TOE1/TOE2 in the CK pathway and supports the idea that CK depends on additional factors in regulating leaf shape.

## Discussion

The regulation of VPC by CK and GA shows striking similarities as both hormones act independently of miR156 and do not affect *SPL* expression, but they both depend on SPL activity as well as positively regulate miR172 levels[6,20,54–56]. Our present study revealed that CK depends on GA biosynthesis and signaling to regulate different aspects of VPC, establishing an epistatic relationship. The results shown here suggest that CK exerts its effect on epidermal identity and leaf shape in

partially different ways and that there might be several points of crosstalk between the two hormones, as depicted in Fig. 6.

Firstly, CK regulates GA metabolism. We found that CK induces GA biosynthesis genes in young plants (Fig. 3). ChIP-seq data shows that the type-B ARRs ARR1, ARR10 and/or ARR12 bind to the *GA1*, *GA2*, *GA20ox2*, and *GA3ox1* loci[69,70], all of which were induced by CK and downregulated in CK-deficient plants (Fig. 3a–c, e–g, i). *GA3ox1* is also included in the "golden list" of genes that are stably upregulated by CK across multiple conditions[71]. These observations indicate a direct regulation of GA biosynthesis genes by type-B ARRs involved in phase change control[20] (Fig. 6). Interestingly, GA3ox1 and GA3ox2 produce the main portion of $GA_4$ in *Arabidopsis*, which is the primary bioactive GA in this species[63,72], and we found $GA_4$ to be less abundant in CK-deficient genotypes (Fig. 4). Together with the finding that enhanced CK signaling has no impact on juvenile leaf number or leaf shape in the background of *ga3ox1 ga3ox2* (Fig. 1c, d), it can be concluded that GA biosynthesis is absolutely required for CK to regulate VPC (Fig. 6). Notably, there is neither a misregulation of GA biosynthesis genes nor changes in $GA_4$ levels in MIM156 or 35S:MIR156 plants[54], underlining the independence of CK and GA from miR156 in this process.

Secondly, CK acts through the GA signaling repressors GAI and RGA, which are both necessary and sufficient to mediate its effect on epidermal identity. Although there is a clear epistatic relationship between the DELLAs GAI and RGA and the miR172 targets TOE1 and

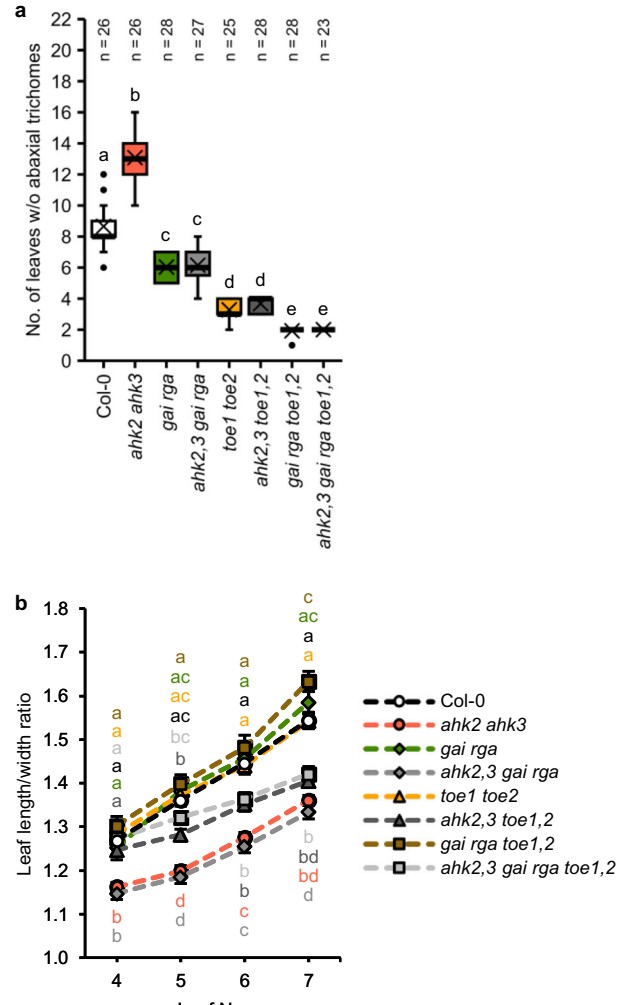

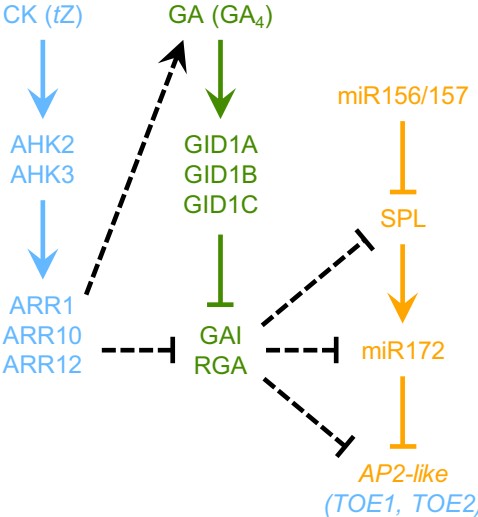

**Fig. 6 | Model for the promotion of vegetative phase change by CK and GA.** The influence of CK, specifically of root-derived *trans*-zeatin (*tZ*), on the regulation of abaxial trichome appearance depends on the signaling components AHK2, AHK3, ARR1, ARR10 and ARR12[20]. These factors are also involved in leaf shape control. CK positively regulates GA biosynthesis including the formation of bioactive GA$_4$, affecting both epidermal identity and leaf shape. The involvement of distinct GA receptors remains to be clarified. The DELLA proteins GAI and RGA are necessary and sufficient to transmit the influence of CK on epidermal identity, whereas leaf shape regulation might involve additional DELLAs. Both CK and GA were shown to positively regulate *MIR172* gene expression[20,54]. In case of epidermal identity, this might be achieved by affecting SPL activity, explaining the dependence on SPLs in this process. The rather minor role of SPLs in leaf shape control by CK indicates a different mode of CK action. Our data suggest a requirement of *AP2-like* genes, which have not previously been implicated in the regulation of this trait. Dotted lines represent hypotheses of pathway interactions based on results of this work combined with previously published data.

**Fig. 5 | The DELLA proteins GAI and RGA are required for the CK-dependent regulation of vegetative phase change. a** Number of leaves without abaxial trichomes of SD-grown *ahk2,3 gai rga*, *ahk2,3 toe1,2*, and *ahk2,3 gai rga toe1,2* hybrid plants in comparison to their respective parents and wild type. In box plots, the center line represents the median value and the boundaries indicate the 25th percentile (upper) and the 75th percentile (lower). The X marks the mean value. Whiskers extend to the largest and smallest value, excluding outliers, which are shown as dots. **b** Length-to-width ratios of the blades of leaves 4 to 7. Data displayed are expressed as mean ± SEM of SD-grown plants. Numbers of biological replicates: Col-0 ($n_4 = 37$; $n_5 = 37$; $n_6 = 38$; $n_7 = 36$), *ahk2 ahk3* ($n_4 = 50$; $n_5 = 50$; $n_6 = 52$; $n_7 = 52$), *gai rga* ($n_4 = 30$; $n_5 = 30$; $n_6 = 31$; $n_7 = 30$), *ahk2,3 gai rga* ($n_4 = 41$; $n_5 = 42$; $n_6 = 42$; $n_7 = 42$), *toe1 toe2* ($n_4 = 35$; $n_5 = 28$; $n_6 = 35$; $n_7 = 34$), *ahk2,3 toe1,2* ($n_4 = 38$; $n_5 = 38$; $n_6 = 39$; $n_7 = 39$), *gai rga toe1,2* ($n_4 = 29$; $n_5 = 28$; $n_6 = 28$; $n_7 = 27$), *ahk2,3 gai rga toe1,2* ($n_4 = 52$; $n_5 = 53$; $n_6 = 53$; $n_7 = 49$). Letters indicate statistically significant differences between the genotypes, as calculated by Kruskal-Wallis test ($q < 0.05$) (**a**) or one-way ANOVA, post-hoc Tukey's test ($p < 0.05$) (**b**).

TOE2 in a CK-deficient background, the even lower juvenile leaf number of *gai rga toe1,2* compared to *toe1 toe2* plants suggests that GA regulates the juvenile-to-adult phase transition also independently of CK (Fig. 5). This might include the remaining DELLA proteins RGL1, RGL2 and RGL3, as well as additional miR172 target genes. Furthermore, some GA-mediated effects on the timing of VPC might not be dependent on the miR172/AP2-like module at all, but might rather be directly carried out by SPLs.

While CK exerts its effect on epidermal identity in dependence of SPLs, GAI/RGA and TOE1/TOE2, the age-dependent changes of leaf shape are regulated differently. Enhanced CK signaling still promotes elongation of the leaf blade in plants lacking functional SPL2/9/10/11/

13/15 (Supplementary Fig. 6d), whereas its effect on epidermal identity is completely blocked (Supplementary Fig. 6c). Other miR156-targeted *SPL* genes, namely *SPL3*, *SPL4*, *SPL5*, and *SPL6*, do not have major roles in VPC regulation[9]. Hence, CK may not rely to a great degree on SPL activity in this process. Instead, our study implicates miR172-targeted *AP2-like* genes to regulate leaf shape in a CK-dependent manner. This is suggested by the observation that lack of *TOE1* and *TOE2* results in elongation of the leaf blade in a CK-deficient background (Fig. 5b). Although the leaf length-to-width ratio of *gai rga* and *ahk2,3 gai rga* does not differ from wild type and *ahk2 ahk3*, respectively (Fig. 5b), CK action might still generally involve DELLA proteins or GA signaling. This is suggested by the fact that enhanced CK signaling fails to increase the leaf length-to-width ratio in a GA-deficient background (Fig. 1b, d). In addition, it was shown that GA metabolism affects the expression of GA signaling components[44,51,73], and *ga1* and *gid1* mutants are very similar on the phenotypic level as well as on the level of gene expression[45].

The aligned activities of CK and GA in VPC control contrast the antagonistic interaction between the two hormones that was demonstrated for numerous processes, including shoot and root elongation, cell differentiation, female-germline cell specification, shoot regeneration in culture, and meristem activity[57–59]. The CK-GA crosstalk in these processes is mediated by various proteins, such as KNOTTED1-like homeobox (KNOXI) family members, the energy sensor TARGET OF RAPAMYCIN (TOR), or the *O*-linked *N*-acetylglucosamine (*O*-GlcNAc) transferases (OGTs) SPINDLY (SPY) and SECRET AGENT (SEC). KNOXI proteins maintain normal shoot apical meristem (SAM) function by controlling the relative levels of CK and GA. They induce CK production, whereas biosynthesis of GA is repressed, and both KNOXI

and CK induce GA breakdown at the base of the SAM, probably in order to prevent biologically active GA from entering the SAM via transport[58,74–77]. In tomato, CK and GA antagonistically affect the activity of the protein kinase TOR, which is an important regulatory hub, playing an important role in the regulation of cellular, developmental, and plant immunity processes[78]. In *Arabidopsis*, SPY and SEC attach single sugar moieties to serine or threonine residues on a number of regulatory proteins, including the GA signaling repressing DELLA proteins[79–81] and the type-B regulator ARR1[82]. *Spy* mutants display a wide range of developmental defects, most of which are attributed to enhanced GA signaling[80,83–85]. In the absence of GA, SPY represses GA signaling by increasing DELLA activity and promotes CK responses. But in case of high GA levels, SPY activity, and therefore CK responses, are inhibited[57,83], indicating that SPY balances the antagonistic activities of the two hormones.

In agreement with the high GA status in the *spy* mutants, abaxial trichomes occur significantly earlier than in wild type[10,86,87]. In contrast, loss of *SEC* does not alter juvenile leaf number, but reduces the number of juvenile leaves of *spl9 spl15* mutants, indicating that *SEC* nevertheless plays a role in timing of VPC. Both single mutants exhibit no changes in miR156 abundance, but higher miR172 levels. Furthermore, SPY interacts with and modifies SPL15, suggesting that SPY inhibits SPL activity by glycosylation[86,87]. Although these observations concur with the role of GA in VPC control, the low CK signaling in *spy* mutants seems like a contradiction. However, as GA acts downstream of the CK pathway in VPC control as was shown here, this might abbreviate the need for CK signaling, providing an explanation for the CK independence of the *spy* phenotype.

Besides CK and GA, several other hormones have been shown to modify the timing of the transition from the juvenile to the adult vegetative phase, including abscisic acid (ABA)[30,88], auxin[6,89,90], brassinosteroids (BR)[29,91], and jasmonic acid (JA)[92–94]. The involvement of SPLs in most of these cases supports the idea that these act as a hub integrating a variety of factors regulating the juvenile-to-adult phase transition. Interaction of SPLs with different co-acting TFs of other pathways is one option. Examples for modulators of SPL activity are JA ZIM-domain (JAZ) proteins, which are the repressors of JA signaling, and BRASSINAZOLE-RESISTANT1 (BZR1), the master transcription factor of the BR signaling pathway[91,94]. The type-B ARRs ARR1, ARR2, ARR10 and ARR12, as well as all of the DELLA repressors bind several SPL proteins involved in VPC regulation, namely SPL2, SPL9, SPL10 and SPL11[54,56]. Furthermore, GAI and RGA were shown to interact with type-B response regulators[95]. It is possible that SPLs, DELLAs and type-B ARRs form a regulatory complex integrating hormonal and possibly other cues (Fig. 6). The output of such a regulatory node could be dynamic and highly context-specific, since the SPL-ARR interaction was shown to dampen CK signaling output instead of enhancing it and GAI and RGA increase the transactivation ability of ARR1 rather than inhibiting it[56,95]. Furthermore, our study shows that the CK-dependent regulation of epidermal identity and leaf shape involves partly different age- and GA-related factors, which implies that the influence of CK on the manifestation of VPC is differential. Consistently, a recent comparison of VPC in 70 *Arabidopsis* accessions found that abaxial trichome production and leaf shape are frequently temporally and genetically uncoupled. This indicated that each of the traits and its temporal expression pattern is regulated at least in part by trait-specific genes, which remain to be discovered[96]. Together, this shows that age pathway regulation is more complex than thought previously.

In summary, we found an epistatic relationship between CK and GA in regulating VPC, in which CK positively regulates GA biosynthesis and signaling. How CK exerts its influence on the different aspects of VPC and whether DELLA activities are directly suppressed by interactions with type-B ARRs and/or as a consequence of elevated GA biosynthesis activating the GA signaling cascade remains to be resolved. The GA receptors mediating the influence of CK on VPC still have to be identified as well. Despite these open questions, our work reveals an agonistic rather than antagonistic interaction between the two hormones, expanding our understanding of VPC regulation and phytohormone crosstalk in general.

## Methods

### Plant material and growth conditions

The Columbia-0 (Col-0) ecotype of *Arabidopsis thaliana* was used as the wild type. All mutants and transgenic lines that were used in this study and generated by crossings are listed in Supplementary Table 2. In the *gai* mutant used in this study has not been fully characterized previously and corresponds to SAIL_587_C02[97]. No full-length *GAI* transcript was detected (Supplementary Fig. 7a), suggesting that this is a null allele. The T-DNA is located at bp 272 (Supplementary Fig. 7b). All genotypes were propagated under LD conditions (16 h dark/8 h light cycle), 22 °C and 30–65 % humidity, and confirmed by PCR analysis. Primers used for genotyping are listed in Supplementary Table 3. For germination of the *ga1* mutant, seeds were incubated in a GA solution (100 μM GA$_3$/100 μM GA$_{4+7}$/0.01 % Tween-20) for three days, washed eight times with water and then transferred to soil. For the analyses of juvenile leaf number and leaf shape as well as gene expression analysis in CK mutants, *Arabidopsis* plants were grown on soil with a 8 h light/16 h dark cycle, at 22 °C and 60 % humidity and light intensities of 100–150 μmol m$^{-2}$ s$^{-1}$. SD conditions were chosen because phenotypic differences are more pronounced than under long days[20]. Pots of different lines were randomized by default to minimize positional effects. For the analysis of *GAI* knock-out in the *gai* mutant, plants were grown on ½ MS agar plates (0.22 % (w/v) MS basal salt, 0.05 % (w/v) MES, 0.5 % (w/v) sucrose, 0.8 % (w/v) agar, pH 5.7) for 10 days under LD conditions.

### Phenotypic analyses

The onset of abaxial trichome formation was scored using a stereomicroscope. For leaf shape analysis, fully expanded leaves were removed, flattened using sticky tape and glass plates, and photographed with a Nikon D3300 camera (Nikon Corp., Tokyo, Japan). Leaf images were measured using ImageJ.

### CK induction assay

For the determination of the influence of CK on GA gene expression, seeds were surface-sterilized using a 1.2 % (v/v) sodium hypochlorite/0.01 % (v/v) Triton X-100 solution. Seedlings were grown under SD conditions for 10 d in liquid ½ MS medium (0.22 % (w/v) MS basal salt, 0.05 % (w/v) MES, 0.1 % (w/v) sucrose, pH adjusted to 5.7). 6-Benzylaminopurine (BA) was dissolved in 1 M KOH. As a control, 1 M KOH was used. Both solutions were diluted in 0.05 % (w/v) MES and the pH was adjusted before adding them to the medium. CK application and harvesting of plant material at the different time points was conducted during the night, starting 1.5 h after its beginning. Successful induction of the CK response was verified by detecting type-A *ARR* transcript levels as CK marker genes[98] via qRT-PCR.

### GA treatment assay

The spraying solutions contained 100 μM GA$_3$, 100 μM GA$_{4+7}$ or H$_2$O as well as 0.01 % Tween-20. Plants were grown under SD conditions and sprayed every other day, starting at 3 DAG.

### RNA preparation and quantitative RT-PCR

Approximately 100 mg of plant material was harvested and frozen in liquid nitrogen at the indicated time points. The frozen samples were ground using a Retsch mill in precooled adapters. Total RNA was extracted using TRIsure™ (Bioline) following the manufacturer's instructions. 80 % (v/v) ethanol was used to wash the RNA pellet, which was resuspended in 40-50 μl of nuclease-free water and treated with DNase I (ThermoFisher) following the manufacturer's instructions. For cDNA synthesis, 1–1.5 μg of total RNA was reversely transcribed using

SuperScript™ III (ThermoFisher), 4.5 μM of N9 random oligos and 2.5 μM of oligo-dT$_{25}$ in a 20 μl reaction. Mix 1 containing RNA, 2 mM of dNTP mix and oligos was incubated for 5 min at 65 °C and placed on ice afterwards. Mix 2 (first strand buffer, 5 mM DTT, SuperScript™ III) was added and samples were incubated for 30 min at 25 °C, 60 min at 50 °C and 15 min at 70 °C. The resulting cDNA was diluted 1:5. For qRT-PCR analyses, *PROTEIN PHOSPHATASE 2A SUBUNIT A2 (PP2AA2)* and *TBP-ASSOCIATED FACTOR II 15 (TAFII15)* served as reference genes. All qRT-PCR primers used in this study are listed in Supplementary Table 4. qRT-PCR was performed with the CFX96™ Real-Time Touch System (Bio-Rad®) using SYBR Green I as DNA-binding dye. Gene expression data analysis was carried out according to Vandesompele et al.[99].

## GA metabolite measurements

The sample preparation and analysis of GAs was performed according to the method described in Urbanová et al.[100] with some modifications. Briefly, whole shoots of SD-grown plants (n = 5) with an average weight of approximately 35 mg fresh weight were harvested at the indicated time points, immediately frozen in liquid nitrogen and stored at −70 °C until analysis. Then the plant material was ground to a fine consistency using 2.7 mm ceria stabilized zirconium oxide beads (Next Advance Inc., Averill Park, NY, USA) and an MM 400 vibration mill at a frequency of 27 Hz for 3 min (Retsch GmbH & Co. KG, Haan, Germany) with 1 ml of ice-cold 80 % acetonitrile containing 5 % formic acid as extraction solution. The samples were then extracted overnight at 4 °C using a benchtop laboratory rotator Stuart SB3 (Bibby Scientific Ltd., Staffordshire, UK) after adding internal GA standards ([$^2H_2$]GA$_1$, [$^2H_2$]GA$_4$, [$^2H_2$]GA$_9$, [$^2H_2$]GA$_{19}$, [$^2H_2$]GA$_{20}$, [$^2H_2$]GA$_{24}$, [$^2H_2$]GA$_{29}$, [$^2H_2$]GA$_{34}$ and [$^2H_2$]GA$_{44}$) purchased from OlChemIm, Czech Republic. The homogenates were centrifuged at 36670 g and 4 °C for 10 min (Hermle Z 35 HK, Hermle Labortechnik GmbH, Germany), and corresponding supernatants were further purified using mixed-mode SPE cartridges (Oasis® MAX, 60 mg/3 ml; Waters, Ireland) using a protocol described in Urbanová et al.[100]. After SPE purification, the samples were evaporated to dryness *in vacuo* (CentriVap® Acid-Resistant benchtop concentrator, Labconco Corp., USA), reconstructed in mobile phase (MeOH: 10 mM formic acid, 1:9 (v/v)) and analyzed by ultra-high performance liquid chromatography-tandem mass spectrometry (UHPLC-MS/MS) using an Acquity UPLC I-Class Plus system (Waters, USA) coupled to a triple quadrupole mass spectrometer Xevo TQ-XS (Waters, USA). The MS settings were as follows: capillary voltage 1.5 kV, cone voltage 30 V, source temperature 150 °C, desolvation gas temperature 600 °C, cone gas flow 150 l/h and desolvation gas flow 1000 l/h. GAs were detected using multiple-reaction monitoring mode of the transition of the ion [M−H]$^-$ to the appropriate product ion (for settings of individual transitions see Urbanová et al.[100]. Masslynx 4.2 software (Waters, Milford, MA, USA) was used to analyze the data, and the standard isotope dilution method[101] was used to quantify the GA levels.

## Statistical analysis

Statistical analyses were performed using GraphPad Prism, version 9 (GraphPad Software, La Jolla, CA). Statistical tests used were all two-sided and are indicated in the figure and table legends. Normally distributed data was analyzed by one-way analysis of variance (ANOVA) followed by Dunnett's post hoc test, or two-way ANOVA followed by Tukey's test. For nonparametric statistics Kruskal-Wallis test followed by two-stage step-up procedure of Benjamini, Krieger and Yuketieli, or two-tailed Mann-Whitney test were used. A *p*- or *q*-value < 0.05 was considered to indicate a statistically significant difference. In case of transcript analyses, a 1.5-fold up- or down-regulation compared to the respective control was chosen as threshold.

## Reporting summary

Further information on research design is available in the Nature Portfolio Reporting Summary linked to this article.

## Data availability

The original contributions presented in the study are included in the article and in the supplementary material. Source data and statistical information are provided with this paper.

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

## Acknowledgements

We acknowledge excellent technical assistance by Gabriele Grüschow, Mgr. Renáta Plotzová and student helper Vera Grießer, and thank Lea Mae Hagelstein and Till Reiß for their contributions made during their bachelor theses. We thank the Nottingham Arabidopsis Stock Center for providing mutant seeds. D.T acknowledges funding provided by the European Regional Development Fund Project "Towards Next Generation Crops" (No. CZ.02.01.01/00/22_008/0004581).

## Author contributions

S.W. and T.S. developed the project; S.W. and D.T. performed experiments; S.W., D.T., and T.S. analyzed data; S.W. and T.S. wrote the article. All authors read and contributed to previous versions and approved the final version.

## Funding

## Competing interests

The authors declare no competing interest.
