## [Transparent Peer Review file · Nature Communications]

Cytokinin depends on GA biosynthesis and signaling to regulate different aspects of vegetative phase change in Arabidopsis

Corresponding Author: Professor Thomas Schmulling

Version 0:

Reviewer comments:

Reviewer #1

(Remarks to the Author)

The manuscript titled "Interplay between Cytokinin and GA in Regulating Vegetative Phase Change in Arabidopsis" by Werner et al. investigates the relationship between cytokinin and GA in the regulation of vegetative phase change. This study builds upon previous research titled "Cytokinin Regulates Vegetative Phase Change in Arabidopsis thaliana through the miR172/TOE1-TOE2 Module." In their earlier work, the authors demonstrated that two miR172 targets, TARGET OF EAT1 (TOE1) and TOE2, encoding transcriptional repressors, are essential for mediating the influence of cytokinin on vegetative phase change.

Vegetative phase change involves a decline in miR156/miR157 levels and an increase in miR156/7 targeted SPLs. This transition is marked by the appearance of trichomes on the abaxial side of leaves, as well as changes in leaf shape and margin serration. SPLs activate miR172, which targets TOE1, TOE2, and TOE3. These TOE proteins interact with leaf polarity genes encoding KAN proteins, leading to the repression of trichome initiation genes such as GLABRA1. Overexpression of miR172 (35S::miR172) can largely restore abaxial trichome production in 35S::miR156 plants, but not the lamina shape, indicating that the miR172-TOE pathway primarily regulates abaxial trichome production in vegetative leaves. However, the authors of the current manuscript only examined the number of leaves without abaxial trichomes and the involvement of the miR172 and TOE pathways. Their findings suggest that the cytokinin and GA pathways control trichome development in vegetative leaves rather than vegetative phase change. Furthermore, previous research by Gan et al. (2007) has already elucidated the involvement of CK and GA in trichome development.

Figure 3 indicates that GA biosynthesis genes are downregulated in cytokinin-deficient mutants or upregulated when plants are treated with CK. However, it is unclear if CK directly activates GA biosynthesis genes. Additionally, the concentrations of total GA or bioactive forms of GA were not altered in any of the CK-deficient plants at all examined time points. Although the concentration of GA4 appears lower than that of the Col control in CKX1ox at day 14, this trend is not observed in the ahk2 ahk3 double mutant, which exhibits similar behavior as CKX1ox in GA biosynthesis. Interestingly, the concentration of GA4 appears lower than that of the Col control in the ahk2 ahk3 double mutant at day 21, but not in CKX1ox at day 21. This inconsistency does not support the assertion that "CK promotes GA biosynthesis" (line 128).

Regarding lines 179-180, the results from the rock2 gid1b,c triple mutant and rga ahk2 ahk3 compared to gid1b,c and rga respectively do not support the claim that "The DELLA genes GAI and RGA are necessary and sufficient to mediate the influence of CK on vegetative phase change." If GA and RGA were necessary and sufficient for mediating the influence of CK on vegetative phase change, the triple mutants should exhibit similar phenotypes to the gid1b,c mutant or rga mutant. However, the results did not support this hypothesis.

Reviewer #2

(Remarks to the Author)

Werner and Colleagues studied genetic interactions which reveal that GA action and biosynthesis is required for cytokinins to mediate the juvenile to adult phase transition. Cytokinins also promote active GA synthesis. This is backed up by GA measurements.

Overall, the conclusions on each of the experiments are well supported by the evidence. The conclusion here is the model of a linear pathway where cytokinins are hypostatic. This contributes to our understanding of the cross talk between cytokinins and gibberellins during shoot development

Concerns:

-use of "cytokinin deficient". This term is also used for ahk mutant combinations, which perhaps could be better referred to as CK signalling mutants.

Line 187, rock gid interactions

-As the rock2 gida mutant combination was not retrieved and studied the conclusion here implicating GIDA in the process seems rather strong. Why does GIDA only appears essential in the rock2 background ?

-effect of GA signalling on cytokinin biosynthesis or signalling in this system. Albeit very unlikely from all the interactions known in the literature , in case rga gai in the ahk2,3 background mutant alleles would upregulate CK content or signalling via CRE1 in the ahk2,3 background then this feedback would complicate the model. One would not expect this on the basis of the known interactions between CK and DELLAs. One would rather expect a lower ck content in the della mutants and reduced expression of DELLA-RR targets, but was it measured in this particular mutant combination ?

Reviewer #3

(Remarks to the Author)

In this manuscript, authors explored epistatic relationship between CK and GA in the context of vegetative phase transition (VPC) in Arabidopsis. The major conclusion is that CK signaling is hypostatic to GA signaling. While this finding is novel, the direct connection between CK signaling components and GA biosynthesis remains unclear. The paper is well-written in general and most of the data supported the conclusion. I have several comments that could potentially strengthen the manuscript and solidify the conclusions:

- 1) In Figs 1 and 2, evaluation of the VPC should not be limited solely to the presence of abaxial trichomes; additional leaf traits such as leaf shape and serrations should also be considered. To ensure a comprehensive analysis and draw robust conclusions, the authors should measure these additional traits as well as the levels of miR156, SPLs, and miR172 in both the wild type and mutants. This same approach should be applied in Fig. 5, where the authors explore the involvement of DELLA proteins in the VPC process.
- 2) Fig. 6. Authors stated that "The DELLA proteins GAI and RGA are necessary and sufficient to transmit the influence of CK on vegetative phase change" in the legend. However, in the figure, it seems that ARR1, ARR10 and ARR12 could directly affect the VPC through SPL, thereby bypassing the GA pathway. Notably, it also contradicts the results in Fig. 1 "The vegetative phase change phenotype of GA biosynthesis mutants is not rescued by enhanced CK signaling". This point should be clarified.
- 3) Some important references are missing: For example, a recent review paper on vegetative phase transition has been published in Dev Cell (PMID: 38194910). It has been shown that cell division in the shoot apical meristem is a trigger the declined of miR156 (PMID: 34750273). The role of SPL in leaf serration has been uncovered a decade ago (PMID: 25448000). Additionally, two labs have recently revealed the molecular and cellular mechanism by which SPLs regulate heteroblasty (leaf shape transition) (PMID: 38244542; PMID: 36748203). If authors wanted to frame their work in the VPC, these references should be cited.
- 4) The reason why all the experiments were carried out in short days should be given.

Reviewer #4

(Remarks to the Author)

CK and GA, two main phytohormones, have been previously both reported to positively regulate vegetative phase change in contrast to the antagonistic nature of CK-GA in other developmental processes such as shoot meristem activity. The manuscript by Werner et. al. establishes an epistatic relationship between CK and GA during vegetative phase change. CK exerts its influence on vegetative phase change depending on GA biosynthesis and signaling. This finding provided a good supplement on understanding of phytohormones crosstalk and plant age pathway. However, to address the crosstalk between CK and GA in vegetative phase change process, more evidence rather than this manuscript are required. In addition, there are some concerns that have to be addressed for supporting the conclusions.

1. In lines 24-25 of the abstract section, this manuscript did not provide any result to support the idea of SPLs acting as a hub to integrate hormone signals, I suggest to remove this sentence. Also, in Fig. 6, dotted lines between ARR/DELLA and SPL are only based on previously published data, but not combined with this work and published data as described in the legend.
2. In the introduction section, the authors used a lot of words to present the function of the miR156/miR157-SPL module and the downstream miR172-AP2-like module during the juvenile-to-adult transition, but I don't think this is the key point of this manuscript, the authors should review phytohormones crosstalk in this section.
3. Using the CK-deficient lines and BA treatment, the authors observed a downregulation of many GA biosynthesis genes including GA1/2, GA20ox1 and GA20ox2, but why the total GA and bioactive GA did not alter? Although the authors claim GA4 may be specifically induced by CK for regulation of vegetative phase change, as the products of GA20oxs, why do the concentrations of GA24 and GA9 appear not to be altered?

4. Results in Fig 5 showed that there is no statistic difference between toe1,2 double mutant and gai rga toe1,2 quadruple mutant (de vs e), but the authors claimed that the quadruple mutant displayed a slight but statistically significant further decrease in juvenile leaf number compared to toe1 toe2 plants (line 205)? The conclusion that GA regulates the juvenile-to-adult phase transition also independently of CK needs a modification.
5. Based on the data in Fig. 1, Fig. 2, and Fig. 5, GA treatment (or GA biosynthesis mutants) and della mutant completely rescued the delayed vegetative phase change of CK-deficient plants, meaning that the function of CK in the regulation of vegetative phase change completely depends on GA, and no other pathway. In addition, SPL transcript levels were not different between ahk2 ahk3 and wild type or after CK treatment (Werner et al, Nature communications 2021), I wonder that why the authors concluded that CK can affect SPL dependently in Fig. 6?
6. More physiological, molecular and genetic evidence have to provide to support the conclusions drawn in this study. For example, employing a physiological process (e.g. shoot meristem development) that indicates the antagonistic nature of CK-GA as a control, to reveal the molecular regulatory difference of CK-GA between two developmental processes (cooperative or antagonistic relationship); the potential role of ARR1s in regulating the expression of GA biosynthesis genes; and the regulation of CK in GA signaling pathway in addition to GA biosynthesis, etc.
7. Please provide the full name of abbreviations, e.g., VPC in line 104 and DAG in line 143;
8. In Fig.1a, there is no * and **, please remove them in line 634.

Version 1:

Reviewer comments:

Reviewer #1

(Remarks to the Author)

The manuscript titled “Cytokinin Depends on GA Biosynthesis and Signaling to Regulate Different Aspects of Vegetative Phase Change in Arabidopsis” is a revised version of the previously submitted “Interplay Between Cytokinin and GA in Regulating Vegetative Phase Change in Arabidopsis” by Werner et al. This study investigates the relationship between cytokinin (CK) and gibberellin (GA) in regulating vegetative phase change (VPC) and concludes that CK functions upstream of GA in this process. While the authors have improved the manuscript, several concerns remain:

The authors argue that assessing VPC based solely on the presence of abaxial trichomes is sufficient. However, two reviewers previously noted that “evaluation of VPC should not be limited solely to the presence of abaxial trichomes; additional leaf traits such as leaf shape and serrations should also be considered.” VPC results from shoot maturation, leading to morphological and physiological differences in emerging leaves. While abaxial trichome production is a marker of VPC, it is not the only one. Genes regulating trichome development may influence abaxial trichomes without affecting leaf shape or margin serration, indicating that VPC itself is not altered. Although the authors have now included leaf blade length-to-width ratios, heteroblasty should be examined across genotypes. I recommend incorporating heteroblasty analyses in Figures 1, 2, 5, S5, and S6 to provide a more comprehensive view of how CK and GA interact to regulate leaf development. Even if previous studies have addressed heteroblasty, it is crucial to evaluate these traits under the same experimental conditions.

Since VPC reflects shoot maturation, factors influencing VPC—whether through the miR156-SPL pathway or otherwise—should exhibit a temporal expression pattern corresponding to shoot maturation. If CK and GA regulate VPC, their biosynthesis or signaling components should display temporal changes, such as increased expression of biosynthetic genes. Providing such evidence would significantly strengthen the manuscript's conclusions.

Lines 252–253: The gai rga toe1 toe2 quadruple mutant produces fewer juvenile leaves than its parental lines, suggesting an additive interaction among these genes. This differs from the genetic interaction between ahk2 ahk3 and toe1 toe2 and does not support the conclusion that CK functions upstream of GA in regulating VPC.

Line 294: The statement that “lack of TOE1 and TOE2 results in elongation of the leaf blade in a CK-deficient background” is unclear. Since toe1 toe2 mutants do not alter the leaf length-to-width ratio, and toe1 toe2 ahk2 ahk3 mutants show a lower ratio than WT or toe1 toe2, this suggests that CK influences leaf blade development through pathways beyond TOE1/TOE2. The wording should be clarified to accurately reflect these findings.

Figure 5b: The overlapping lines make it difficult to distinguish between groups. Converting the line graph into a bar graph with statistical analyses would enhance clarity and help readers determine which groups differ significantly.

Reviewer #2

(Remarks to the Author)

In my opinion, the evidence presented here (including the additional genetic analysis of rock2 spl mutants) support the conclusions and the main message of the manuscript. The evidence involves detailed analysis of genetic interactions in terms of juvenile leaves and leaf shape, transcript levels and GA measurements.

Main findings:

Cytokinin signalling influences the biosynthesis of a subset of bioactive GA

GA signalling mediates epidermal identity.

Cytokinin influences leaf shape through multiple pathways, including SPL independent pathways

The revisions of the manuscript addressed my concerns

Reviewer #3

(Remarks to the Author)

I think authors have addressed all my concerns.

Reviewer #4

(Remarks to the Author)

The authors have provided convincing explanation or correction to address most of my concerns. However, as raised in the last review, I still suggest that the authors provide molecular evidence to demonstrate the potential role of ARR in regulating the expression of GA biosynthesis genes, although ARRs have been shown to bind to these genes according to ChIP-seq data. This experiment is necessary to verify the direct function of ARR regulating its targets, and not difficult for the authors to complete.

REVIEWER COMMENTS

Reviewer #1 (Remarks to the Author):

The manuscript titled "Interplay between Cytokinin and GA in Regulating Vegetative Phase Change in Arabidopsis" by Werner et al. investigates the relationship between cytokinin and GA in the regulation of vegetative phase change. This study builds upon previous research titled "Cytokinin Regulates Vegetative Phase Change in Arabidopsis thaliana through the miR172/TOE1-TOE2 Module." In their earlier work, the authors demonstrated that two miR172 targets, TARGET OF EAT1 (TOE1) and TOE2, encoding transcriptional repressors, are essential for mediating the influence of cytokinin on vegetative phase change.

Vegetative phase change involves a decline in miR156/miR157 levels and an increase in miR156/7 targeted SPLs. This transition is marked by the appearance of trichomes on the abaxial side of leaves, as well as changes in leaf shape and margin serration. SPLs activate miR172, which targets TOE1, TOE2, and TOE3. These TOE proteins interact with leaf polarity genes encoding KAN proteins, leading to the repression of trichome initiation genes such as GLABRA1. Overexpression of miR172 (35S::miR172) can largely restore abaxial trichome production in 35S::miR156 plants, but not the lamina shape, indicating that the miR172-TOE pathway primarily regulates abaxial trichome production in vegetative leaves.

1) However, the authors of the current manuscript only examined the number of leaves without abaxial trichomes and the involvement of the miR172 and TOE pathways. Their findings suggest that the cytokinin and GA pathways control trichome development in vegetative leaves rather than vegetative phase change. Furthermore, previous research by Gan et al. (2007) has already elucidated the involvement of CK and GA in trichome development.

Answer: We mention in our manuscript, that both CK and GA act independently of miR156 (lines 88-91) and that the miR156/SPL and miR172/AP2-like modules exhibit a certain independence, regulating different aspects of VPC, with only the former controlling also other morphological traits than epidermal identity (lines 51 to 56). We therefore focused on appearance of abaxial trichomes, which also has been commonly used to define the transition from the juvenile to the adult vegetative phase in Arabidopsis¹. Furthermore, Wang et al. (2019)² showed that abaxial trichome formation is spatiotemporally regulated by the leaf abaxial identity determinant KAN1 and miR172-targeted AP2-like proteins, including TOE1, repressing abaxial trichome production in the juvenile phase by inhibiting *GLABRA* expression. The increasing miR172 abundance with progressing plant age (and therefore decreasing AP2-like activity) allows *GL1* expression resulting in abaxial trichome appearance. Hence, abaxial trichome formation is directly linked to the CK- and GA-regulated miR172-AP2-like module of the age pathway and therefore part of VPC.

Nevertheless, we understand the request and included the leaf length-to-width ratio for key experiments in the revised version as a second parameter to describe VPC. We already mentioned in our previous study, that also leaf shape is affected by CK, but did not go into detail about this for the above-mentioned reasons. Addressing this additional parameter in the new manuscript actually uncovered some regulatory differences between the CK-dependent control of epidermal identity and leaf shape and thus broadened our understanding of VPC regulation.

Finally, it is true that Gan et al. (2007) already dealt with the involvement of CK and GA in the regulation of trichome production. But firstly, they investigated the molecular mechanisms through which the two hormones initiate trichomes on inflorescence organs, and secondly the authors found that CK and GA regulate this independently of each other, depending on different C2H2 transcription factors (CK: ZFP8 and GIS2; GA: GIS). So no direct interaction of CK and GA pathways is taking place in this process which is distinct from VPC.

2) Figure 3 indicates that GA biosynthesis genes are downregulated in cytokinin-deficient mutants or upregulated when plants are treated with CK. However, it is unclear if CK directly activates GA biosynthesis genes.

Answer: This is true, although a fast induction of gene expression after CK application might indicate a direct activation. Also, we mention ChIP-seq data showing binding of B-ARRs to the loci of these genes (lines 268-272), indicating that the CK effect on GA gene expression might be direct. Apart from that, there is an undeniable influence of CK on GA biosynthesis, so the question if the effect on GA gene expression is direct or indirect is not a key to understand the genetic architecture.

3) Additionally, the concentrations of total GA or bioactive forms of GA were not altered in any of the CK-deficient plants at all examined time points.

Answer: This is true, we mention it ourselves. This is because only the concentrations of a few metabolites are altered, which make up only a small fraction of the total. We added a sentence at the end of the section dealing with GA biosynthesis to indicate this (lines 196-198).

4) Although the concentration of GA₄ appears lower than that of the Col control in CKX1ox at day 14, this trend is not observed in the *ahk2 ahk3* double mutant, which exhibits similar behavior as CKX1ox in GA biosynthesis. Interestingly, the concentration of GA₄ appears lower than that of the Col control in the *ahk2 ahk3* double mutant at day 21, but not in CKX1ox at day 21. This inconsistency does not support the assertion that "CK promotes GA biosynthesis" (line 128).

Answer: You are right, expression data and GA metabolite data are not completely consistent. GA₄ is mainly produced by GA3ox1 and GA3ox2 and while *GA3ox1* expression in CKX1ox and *ahk2,3* is nearly the same, but lower as WT (Fig. 3g), *GA3ox2* is much more downregulated in CKX1ox compared to *ahk2,3* (Fig. 3h). In contrast to *GA3ox1*, *GA3ox2* is not downregulated in the CK-deficient genotypes at the earliest time point (7 DAG) and at this time point, there are also no differences in GA₄ concentration between WT and the CK-deficient genotypes, which might indicate that *GA3ox2* is of greater importance here.

CKX1ox is more CK-deficient than *ahk2,3* (clearly visible by the more severe shoot phenotype). Consistently, reduced GA₄ occurs earlier in CKX1ox than *ahk2,3* (Fig. 4), which might explain the stronger VPC phenotype of CKX1ox. Indeed, GA₄ is not reduced at 21 DAG in CKX1ox anymore, but we see a stronger reduction of the GA₄ degradation product GA₃₄ for CKX1ox than for *ahk2,3* at all time points. Since we are only looking at snapshots during development and there is a constant turnover of metabolites, looking at the end point of the pathway (GA₃₄) might give the best indication. Of course, this alone is no real proof of GA₄ involvement in the CK-dependent regulation of VPC, but the genetic evidence based on the VPC phenotypes of *rock2 ga3ox1,2* (Fig. 1c, d) strongly corroborates the hypothesis.

5) Regarding lines 179-180, the results from the *rock2 gid1b,c* triple mutant and *rga ahk2 ahk3* compared to *gid1b,c* and *rga* respectively do not support the claim that "The DELLA genes GAI and RGA are necessary and sufficient to mediate the influence of CK on vegetative phase change." If GA and RGA were necessary and sufficient for mediating the influence of CK on vegetative phase change, the triple mutants should exhibit similar phenotypes to the *gid1b,c* mutant or *rga* mutant. However, the results did not support this hypothesis.

Answer: The claim that GAI and RGA are necessary and sufficient to mediate the influence of CK on VPC, or at least epidermal identity, is based on the results for *ahk2,3 gai rga*, showing a clear epistatic relationship between AHK2/3 and GAI/RGA (lines 220-222; Fig. 5a). For the GA receptors GID1A, B and C, we did not make a clear statement, since we could only examine *rock2 gid1b,c* (lines 202-209; Fig. S4).

Reviewer #2 (Remarks to the Author):

Werner and Colleagues studied genetic interactions which reveal that GA action and biosynthesis is required for cytokinins to mediate the juvenile to adult phase transition. Cytokinins also promote active GA synthesis. This is backed up by GA measurements.

Overall, the conclusions on each of the experiments are well supported by the evidence. The conclusion here is the model of a linear pathway where cytokinins are hypostatic. This contributes to our understanding of the cross talk between cytokinins and gibberellins during shoot development.

Concerns:

1) use of "cytokinin deficient". This term is also used for *ahk* mutant combinations, which perhaps could be better referred to as CK signalling mutants.

Answer: The term "cytokinin-deficient" is commonly used for lines with a lower CK status in general, including metabolic as well as signaling mutants. We added a sentence in lines 129-130 to indicate this ("plants that are CK-deficient due to lower CK levels or signaling").

2) Line 187, rock gid interactions: As the *rock2 gida* mutant combination was not retrieved and studied the conclusion here implicating GIDA in the process seems rather strong. Why does GIDA only appears essential in the *rock2* background?

Answer: The idea is that since *rock2* doesn't lose its effect on juvenile leaf number in the *gid1b,c* background, GID1A might be enough to transmit the CK signal to downstream effectors. We used the term "indicates" (line 207) to express that we have no proof for this, but this possibility is consistent with the known functional redundancy of the three GA receptors. We modified the section to make this clearer (lines 206-209). Furthermore, we added "Alternatively, CK could also act GID1-independently." (line 209).

3) effect of GA signalling on cytokinin biosynthesis or signalling in this system. Albeit very unlikely from all the interactions known in the literature, in case *rga gai* in the *ahk2,3* background mutant alleles would upregulate CK content or signalling via CRE1 in the *ahk2,3* background then this feedback would complicate the model. One would not expect this on the basis of the known interactions between CK and DELLAs. One would rather expect a lower ck content in the *della* mutants and reduced expression of DELLA-RR targets, but was it measured in this particular mutant combination?

Answer: In the previous study, we showed that CRE1 is not important (or plays only a minor role) in the CK-dependent regulation of VPC³, which is consistent with its very low (or even absent) expression in leaves^{4,5}. Riefler et al. (2006) indeed reported that reduced CK signaling in *ahk2,3* causes a homeostatic response resulting in a higher CK content⁶. However, this would not have any effect on the leaf phenotype of *ahk2,3 gai rga* because of the lack of the AHK2/AHK3 receptors in that line. The fact that *gai rga* has the same juvenile leaf number as *ahk2,3 gai rga* (Fig. 5a) indirectly tells us that there is likely no increase in CK content in the *della* mutants, since the absence or presence of the CK receptors makes no difference.

Reviewer #3 (Remarks to the Author):

In this manuscript, authors explored epistatic relationship between CK and GA in the context of vegetative phase transition (VPC) in Arabidopsis. The major conclusion is that CK signaling is hypostatic to GA signaling. While this finding is novel, the direct connection between CK signaling components and GA biosynthesis remains unclear. The paper is well-written in general and most of the data supported the conclusion. I have several comments that could potentially strengthen the manuscript and solidify the conclusions:

1) In Figs 1 and 2, evaluation of the VPC should not be limited solely to the presence of abaxial trichomes; additional leaf traits such as leaf shape and serrations should also be considered. To ensure a comprehensive analysis and draw robust conclusions, the authors should measure these additional traits as well as the levels of miR156, SPLs, and miR172 in both the wild type and mutants. This same approach should be applied in Fig. 5, where the authors explore the involvement of DELLA proteins in the VPC process.

Answer: As outlined in the answer to point 1 of reviewer #1 we have included the leaf length-to-width ratio for key experiments in the revised version as a second parameter. We already mentioned in our previous study, that also leaf shape is affected by CK, but did not go into detail about this for the above-mentioned reasons. Addressing this in the new manuscript actually uncovered some regulatory differences between the CK-dependent control of epidermal control and leaf shape and broadened our understanding of VPC regulation.

We refrained from measuring miR156 and *SPL* expression, since it is largely unaffected by either hormone (lines 88-93). On the other hand, there is evidence that SPL activity participates in the CK- and GA-dependent regulation of VPC (lines 93-97). To investigate the importance of SPLs for VPC control by CK further, we carried out additional experiments and included results from *rock2 spl2,10,11,13,15* and *rock2 spl2,9,10,11,13,15* lines in the revised version of the manuscript (lines 227-241; Fig. S6). These data provide proof of the functional relevance of SPLs and highlight the role of SPL9 which was masked by functional redundancy in a previous analysis³.

We attempted to measure miR172 levels for key experiments in the manuscript. Unfortunately, the differences between genotypes were rather low, so that we could not draw clearly significant conclusions from these data. With regard to the finding that different CK-controlled aspects of VPC are not depending on the same components of GA and age-dependent signaling pathways, the expression data might be

of limited relevance anyway. We based our conclusions on the genetic data we obtained from phenotypic analyses.

2) Fig. 6. Authors stated that “The DELLA proteins GAI and RGA are necessary and sufficient to transmit the influence of CK on vegetative phase change” in the legend. However, in the figure, it seems that ARR1, ARR10 and ARR12 could directly affect the VPC through SPL, thereby bypassing the GA pathway. Notably, it also contradicts the results in Fig. 1 “The vegetative phase change phenotype of GA biosynthesis mutants is not rescued by enhanced CK signaling”. This point should be clarified.

Answer: You are absolutely right. The triangle in the center of the model was supposed to convey the point of crosstalk between all three factors, but we understand the confusion and modified the model in order to make it clearer.

3) Some important references are missing: For example, a recent review paper on vegetative phase transition has been published in *Dev Cell* (PMID: 38194910). It has been shown that cell division in the shoot apical meristem is a trigger the declined of miR156 (PMID: 34750273). The role of SPL in leaf serration has been uncovered a decade ago (PMID: 25448000). Additionally, two labs have recently revealed the molecular and cellular mechanism by which SPLs regulate heteroblasty (leaf shape transition) (PMID: 38244542; PMID: 36748203). If authors wanted to frame their work in the VPC, these references should be cited.

Answer: We have included these references and explain the recently uncovered regulation of heteroblasty by SPLs (lines 57-59, 64-67).

4) The reason why all the experiments were carried out in short days should be given.

Answer: The effect of CK on VPC regulation is not photoperiod-dependent. However, under short day conditions, development is slower and differences become more pronounced. We illustrated the differences between long and short days in the previous publication³:

“[...] we investigated a possible dependence of the phenotype on photoperiod since a long photoperiod stimulates abaxial trichome formation (Chien et al., 1996). Plants were grown under both long-day (LD; 16 h of light, 8 h of darkness) and short-day conditions (SD; 8 h of light, 16 h of darkness). We observed the same positive correlation of the altered CK status and the onset of the adult phase under both photoperiods, with more pronounced phenotypic differences under SD. Genotypes with a higher CK status displayed an earlier transition to the adult phase, whereas a reduced CK status prolonged the juvenile phase (Fig. 1).”

We also added an explanation for the choice of short-day conditions in lines 369-370 of the revised version.

Reviewer #4 (Remarks to the Author):

CK and GA, two main phytohormones, have been previously both reported to positively regulate vegetative phase change in contrast to the antagonistic nature of CK-GA in other developmental processes such as shoot meristem activity. The manuscript by Werner et. al. establishes an epistatic relationship between CK and GA during vegetative phase change. CK exerts its influence on vegetative phase change depending on GA biosynthesis and signaling. This finding provided a good supplement on understanding of phytohormones crosstalk and plant age pathway. However, to address the crosstalk between CK and GA in vegetative phase change process, more evidence rather than this manuscript are required. In addition, there are some concerns that have to be addressed for supporting the conclusions.

1) In lines 24-25 of the abstract section, this manuscript did not provide any result to support the idea of SPLs acting as a hub to integrate hormone signals, I suggest to remove this sentence. Also, in Fig. 6, dotted lines between ARRs/DELLA and SPL are only based on previously published data, but not combined with this work and published data as described in the legend.

Answer: We agree on the first point and rewrote the summary, also in view of added results to the revised version. In the model, we attempted to combine the new results with the previously published ones in order to present a working model/hypothesis, with no claim of completeness. The model was modified due to discrepancies brought up by reviewer #3.

2) In the introduction section, the authors used a lot of words to present the function of the miR156/miR157-SPL module and the downstream miR172-AP2-like module during the juvenile-to-adult transition, but I don't think this is the key point of this manuscript, the authors should review phytohormones crosstalk in this section.

Answer: In the introduction we described the age, CK and GA pathways, their known convergence points as well as the known roles of the phytohormones in regulating VPC. The age pathway was introduced to understand the mechanisms underlying VPC regulation and as a basis for the interplay with other factors. The current knowledge about the roles of CK and GA in regulating the age pathway provides the reader with the necessary information about the starting point of our investigations. CK-GA crosstalk is dealt with in detail in the discussion section to compare and link the newly obtained results with the literature.

3) Using the CK-deficient lines and BA treatment, the authors observed a downregulation of many GA biosynthesis genes including GA1/2, GA20ox1 and GA20ox2, but why the total GA and bioactive GA did not alter?

Answer: There are five *GA20ox* genes, of which *GA20ox1* and *GA20ox2* are the most highly expressed *GA20ox* genes during vegetative and early reproductive development and together with *GA20ox3* constitute the dominant paralogs^{7,8}. There are four *GA3ox* genes in Arabidopsis, we focused on the first two, because they appear to produce the main portion of bioactive GA metabolites and to be the most important ones for proper vegetative growth⁹. The other not tested genes might also contribute and be regulated differently. Importantly, only particular metabolites are altered in their levels, which make up only a small portion of total GA. The GA biosynthesis pathway is not linear and one particular enzyme may act at different points of the pathway¹⁰. Which enzyme acts where in the pathway has not been resolved in all detail.

Furthermore, measuring gene expression and metabolites in a plant only shows the situation at a particular point in time – the plant has been growing for a while already and has initiated measures to counterbalance misregulation of important factors, which is why some metabolites might appear not to be altered, while others cannot be compensated. This makes the result of altered GA₄ levels even more striking. We encounter similar phenomena with different CK metabolites. For example, the concentration of storage metabolites may change drastically and indicate an altered CK content while the concentration of biologically active metabolites shows only small or no changes. Furthermore, we extract mixed tissues and also do not know much about the subcellular distribution of GAs. Therefore, all data should be interpreted with caution. The main conclusions rely on genetic data.

4) Although the authors claim GA₄ may be specifically induced by CK for regulation of vegetative phase change, as the products of GA20oxs, why do the concentrations of GA₂₄ and GA₉ appear not to be altered?

Answer: It is true that expression data and GA metabolite data are not completely consistent. But please note that we are only looking at snapshots of a developing plant and there is a constant turnover of metabolites. That's why looking at the end point of the pathway (GA₃₄) might give the best indication. In our opinion it makes sense that CK positively regulates not only *GA3ox* genes but also *GA20ox* genes, since the latter provide the substrates for the *GA3ox* enzymes.

5) Results in Fig 5 showed that there is no statistic difference between *toe1,2* double mutant and *gai rga toe1,2* quadruple mutant (de vs e), but the authors claimed that the quadruple mutant displayed a slight but statistically significant further decrease in juvenile leaf number compared to *toe1 toe2* plants (line 205)? The conclusion that GA regulates the juvenile-to-adult phase transition also independently of CK needs a modification.

Answer: Looking solely at the statistics, you are absolutely right, but looking at the raw data, you'll only find 2 or 3 leaves with abaxial trichomes for *toe1,2* and *ahk2,3 toe1,2*, whereas *gai rga toe1,2* and *ahk2,3 gai rga toe1,2* only produced 1-2 leaves. Nevertheless, our wording was misleading and should have been more cautious. Since it is hard to get statistically significant results when differences are small, we repeated the experiment once more with slightly lower light intensities, relating to an article published by Xu et al. (2021) showing that low light intensities delay VPC¹¹. As a result, the differences between the genotypes became more distinct and consequently statistically significant. The new figure was included in the revised version of the article.

6) Based on the data in Fig. 1, Fig. 2, and Fig. 5, GA treatment (or GA biosynthesis mutants) and *della* mutant completely rescued the delayed vegetative phase change of CK-deficient plants, meaning that the

function of CK in the regulation of vegetative phase change completely depends on GA, and no other pathway. In addition, SPL transcript levels were not different between *ahk2 ahk3* and wild type or after CK treatment (Werner et al, Nature communications 2021), I wonder that why the authors concluded that CK can affect SPL dependently in Fig. 6?

Answer: Fig. 4c in the previous study shows a cross between the CK gain-of-function mutant *rock2* (constitutively active CK receptor) and a miR156 overexpressor (no SPL activity)³. The *rock2* mutation is not able to accelerate the transition to the adult phase in that background implicating the importance of SPLs in the CK-dependent regulation of VPC. We included an additional supplemental figure in the revised version of the manuscript confirming this result with a *rock2 spl2,9,10,11,13,15* line: In the background of this *SPL* multiple mutant, *rock2* is no longer able to induce an earlier appearance of abaxial trichomes (Fig. S6c, lines 227-241). *SPL* transcript levels are neither affected by CK nor by GA (lines 88-93) and it has been shown that miR156 regulates SPLs primarily by promoting their translational repression¹².

7) More physiological, molecular and genetic evidence have to provide to support the conclusions drawn in this study. For example, employing a physiological process (e.g. shoot meristem development) that indicates the antagonistic nature of CK-GA as a control, to reveal the molecular regulatory difference of CK-GA between two developmental processes (cooperative or antagonistic relationship); the potential role of ARRs in regulating the expression of GA biosynthesis genes; and the regulation of CK in GA signaling pathway in addition to GA biosynthesis, etc.

Answer: To analyze the molecular regulatory differences between cooperative and antagonistic CK-GA crosstalk has not been the goal of our work. Instead, we have focused on elucidating the genetic architecture of the CK/GA pathways in regulating vegetative phase change. The results provide hypotheses for further molecular analyses, e.g. the potential functional relevance of a hypothetical SPL/DELLA/ARR complex that could exist based on molecular interaction data etc., which will be addressed in future work.

Notably, our results do not contradict any previously published data, which might have given a reason to follow this in more detail. In addition, the involvement of the age pathway might already give an explanation for the different mode of crosstalk. In the discussion, we provide several possibilities of CK-GA interactions (positive influence of CK on GA biosynthesis, ARR-DELLA interaction etc.). Although the exact mode of CK action remains elusive, our genetic analysis has identified a number of factors being involved (ARR1,10,12, GAI, RGA, SPLs), therefore indicated by dotted lines in the model.

Type-B ARRs act as transcription factors to mediate the CK signaling output and ChIP-seq data shows that they bind to the gene loci investigated here, indicating a direct CK effect on expression of GA-related genes (lines 267-271). The fast induction of gene expression after CK application might also indicate a direct activation of these genes (Fig. 3c, f, i). These points are supportive for the model, which is based on genetic data. Exploring the exact molecular mechanism by which CK regulates GA biosynthesis will be the objective of future work.

8) Please provide the full name of abbreviations, e.g., VPC in line 104 and DAG in line 143;

Both abbreviations are now introduced in the main text (lines 38 and 162).

9) In Fig.1a, there is no * and **, please remove them in line 634.

*p < 0.01 and **p < 0.05 have been removed from the Fig. 1 legend.

References

1. Telfer, A., Bollman, K. M. & Poethig, R. S. Phase change and the regulation of trichome distribution in *Arabidopsis thaliana*. *Development* **124**, 645-654 (1997).
2. Wang, L., et al. A spatiotemporally regulated transcriptional complex underlies heteroblastic development of leaf hairs in *Arabidopsis thaliana*. *EMBO Journal* **38**, e100063 (2019).
3. Werner, S., Bartrina, I. & Schmülling, T. Cytokinin regulates vegetative phase change in *Arabidopsis thaliana* through the miR172/TOE1-TOE2 module. *Nat. Commun.* **12**, 5816 (2021).
4. Higuchi, M., et al. *In planta* functions of the *Arabidopsis* cytokinin receptor family. *Proc. Natl. Acad. Sci. USA* **101**, 8821-8826 (2004).

5. Nishimura, C., et al. Histidine kinase homologs that act as cytokinin receptors possess overlapping functions in the regulation of shoot and root growth in *Arabidopsis*. *Plant Cell* **16**, 1365-1377 (2004).
6. Riefler, M., Novak, O., Strnad, M. & Schmülling, T. *Arabidopsis* cytokinin receptor mutants reveal functions in shoot growth, leaf senescence, seed size, germination, root development, and cytokinin metabolism. *Plant Cell* **18**, 40-54 (2006).
7. Rieu, I., et al. The gibberellin biosynthetic genes *AtGA20ox1* and *AtGA20ox2* act, partially redundantly, to promote growth and development throughout the *Arabidopsis* life cycle. *Plant Journal* **53**, 488-504 (2008).
8. Plackett, A. R. G., et al. Analysis of the developmental roles of the *Arabidopsis* gibberellin 20-oxidases demonstrates that GA20ox1, -2, and -3 are the dominant paralogs. *Plant Cell* **24**, 941-960 (2012).
9. Mitchum, M. G., et al. Distinct and overlapping roles of two gibberellin 3-oxidases in *Arabidopsis* development. *Plant Journal* **45**, 804-818 (2006).
10. Yamaguchi, S. Gibberellin metabolism and its regulation. *Annu. Rev. Plant Biol.* **59**, 225-251 (2008).
11. Xu, M., Hu, T. & Poethig, R. S. Low light intensity delays vegetative phase change. *Plant Physiology* **187**, 1177-1188 (2021).
12. Xu, M., et al. Developmental functions of miR156-regulated SQUAMOSA PROMOTER BINDING PROTEIN-LIKE (SPL) genes in *Arabidopsis thaliana*. *PLoS Genetics* **12**, e1006263 (2016).

NCOMMS-24-10528A

Werner et al.: Cytokinin depends on GA biosynthesis and signaling to regulate different aspects of vegetative phase change in *Arabidopsis*

Reviewer #1 (Remarks to the Author):

The manuscript titled “Cytokinin Depends on GA Biosynthesis and Signaling to Regulate Different Aspects of Vegetative Phase Change in *Arabidopsis*” is a revised version of the previously submitted “Interplay Between Cytokinin and GA in Regulating Vegetative Phase Change in *Arabidopsis*” by Werner et al. This study investigates the relationship between cytokinin (CK) and gibberellin (GA) in regulating vegetative phase change (VPC) and concludes that CK functions upstream of GA in this process. While the authors have improved the manuscript, several concerns remain:

1) The authors argue that assessing VPC based solely on the presence of abaxial trichomes is sufficient. However, two reviewers previously noted that “evaluation of VPC should not be limited solely to the presence of abaxial trichomes; additional leaf traits such as leaf shape and serrations should also be considered.” VPC results from shoot maturation, leading to morphological and physiological differences in emerging leaves. While abaxial trichome production is a marker of VPC, it is not the only one. Genes regulating trichome development may influence abaxial trichomes without affecting leaf shape or margin serration, indicating that VPC itself is not altered. Although the authors have now included leaf blade length-to-width ratios, heteroblasty should be examined across genotypes. I recommend incorporating heteroblasty analyses in Figures 1, 2, 5, S5, and S6 to provide a more comprehensive view of how CK and GA interact to regulate leaf development. Even if previous studies have addressed heteroblasty, it is crucial to evaluate these traits under the same experimental conditions.

Answer: First, we would like to clarify that we do not claim that assessing VPC solely based on abaxial trichome appearance is sufficient. But since CK only impacts miR172 abundance, and not miR156 abundance, and because the miR172/AP2-like module was not known so far to impact other heteroblastic features besides epidermal identity, we focused on epidermal identity to define the timing of VPC in the previous study¹ and the first version of the present study. The functions and certain independence of the two modules of the age pathway are described in detail in the introduction (lines 37-67). We also explain the role of CK and the reasons for using the epidermal identity parameter at first (lines 68-70, 88-93, 117-119).

That said, we absolutely understood the request for incorporating a second parameter and gladly complied. Therefore, we have analyzed for the revised version the leaf length-to-width ratio for all the main experiments (Figs. 1, 2 and 5), as well as Supplemental Figure 3. We refrained from doing the same for the genotypes shown in Supplementary Figures 4 and 5 for the following reasons: Supplemental Figure 4 is not fully conclusive concerning the role of the GA receptors in VPC regulation in general. The significance of the genotypes shown in Supplementary Figure 5 would be low and the work needed would in our opinion not be justified because of functional redundancy among DELLA genes in general as well as the results of the higher order della mutants shown in Fig. 5.

In sum, we think that we have addressed heteroblasty sufficiently by analyzing the two parameters mostly used in studies dealing with VPC (lines 64-65).

2) Since VPC reflects shoot maturation, factors influencing VPC—whether through the miR156-SPL pathway or otherwise—should exhibit a temporal expression pattern corresponding to shoot maturation. If CK and GA regulate VPC, their biosynthesis or signaling components should display temporal changes, such as increased expression of biosynthetic genes. Providing such evidence would significantly strengthen the manuscript's conclusions.

Answer: The temporal expression patterns of the age pathway components are well known and the relevant literature is cited (lines 40-44, 48-51). We showed the influence of CK on the expression of miR156 and miR172 in our previous study¹. The influence of GA is also known and mentioned in the text (lines 88-91). Both hormones do not affect *SPL* expression (lines 91-93). *SPL* participation in the CK- and GA-dependent regulation of VPC has been suggested to take place on the post-translational level instead (lines 93-99). Therefore, we refrained from performing an *SPL* expression analysis. Furthermore, AP2-like factors are also known to rather be regulated on the translational level^{2,3,4}.

As mentioned in the answer to Reviewer #3 in the last review, we attempted to measure miR172 levels for key experiments for the revised version. Unfortunately, the differences between genotypes were rather low, so that we could not draw clear conclusions from these data. With regard to the finding that different CK-controlled aspects of VPC are not depending on the same components of GA and age-dependent signaling pathways, the expression data might be of limited relevance anyway. We have

based our conclusions on the genetic data we obtained from phenotypic analyses and think that these are persuasive.

Importantly, in Fig. 3 and Supplemental Figures 1 and 2, we show the temporal expression patterns of GA metabolism genes in wild type as well as the CK-deficient genotypes CKX1ox and *ahk2 ahk3*. The consequences of these expression patterns for the GA content are also shown (Fig. 4, Supplementary Figure 3, Supplementary Table 1).

Since we describe in our manuscript that CK depends on GA in regulating VPC, studying the expression patterns of CK metabolism genes in wild type is in our opinion not very informative. More generally, we think that correlations between expression levels of synthesis genes of hormones and hormonal activity need to be interpreted with caution and might not give unequivocal information. Changes in spatial expression pattern of genes might be more relevant than changes in expression level, relevant regulation of enzyme activity might occur post-transcriptionally, hormone sensitivity and signaling need to be considered as well etc. In the present case it is important to consider that a relevant part of the cytokinin comes from the roots, which makes it more difficult to identify functionally relevant changes. Taken together, in view of numerous published data on gene expression in the context of vegetative phase change and the limitations of correlating changes in gene expression with the functional relevance of the gene products, we think that additional data on the expression of CK and GA synthesis genes are not needed to support the central message of the manuscript.

3) Lines 252–253: The *gai rga toe1 toe2* quadruple mutant produces fewer juvenile leaves than its parental lines, suggesting an additive interaction among these genes. This differs from the genetic interaction between *ahk2 ahk3* and *toe1 toe2* and does not support the conclusion that CK functions upstream of GA in regulating VPC.

Answer: We agree with the opinion that the interaction between GAI/RGA and TOE1/TOE2 in regulating epidermal identity is additive, which we mention ourselves in the text (lines 252-253). However, there is a clear epistatic relationship between GAI/RGA, TOE1/TOE2 and AHK2/AHK3 as shown in Fig. 5a: *ahk2,3 gai rga* has the same phenotype as *gai rga*, *ahk2,3 toe1,2* has the same phenotype as *toe1 toe2* and *ahk2,3 gai rga toe1,2* has the same phenotype as *gai rga toe1,2*. Therefore, we drew the conclusion that GA also acts independently of CK and “that additional *DELLA* and/or *AP2-like* genes participate in VPC control by GA” (lines 252-253). See also the additional comment on this point in the Discussion (lines 280-286). We think it is conceivable that not only CK acts through GA but that other factors regulate leaf shape through this pathway as well.

4) Line 294: The statement that “lack of TOE1 and TOE2 results in elongation of the leaf blade in a CK-deficient background” is unclear. Since *toe1 toe2* mutants do not alter the leaf length-to-width ratio, and *toe1 toe2 ahk2 ahk3* mutants show a lower ratio than WT or *toe1 toe2*, this suggests that CK influences leaf blade development through pathways beyond TOE1/TOE2. The wording should be clarified to accurately reflect these findings.

Answer: We agree on this point and mention it in the text (lines 244-249, 292-295). There is no clear epistatic relationship between AHK2/AHK3 and TOE1/TOE2 with respect to leaf shape regulation, so other factors are definitively involved. However, the fact that lack of TOE1/TOE2 does only influence leaf shape in the CK-deficient background clearly indicates a link between CK and these two transcriptional regulators in regulating this feature.

5) Figure 5b: The overlapping lines make it difficult to distinguish between groups. Converting the line graph into a bar graph with statistical analyses would enhance clarity and help readers determine which groups differ significantly.

Answer: We have changed the graph into a bar chart as suggested and show one version below. However, we did not find the result convincing, as different groups are in our opinion not easier to distinguish than in the line graph. As an alternative, we modified the previous version of the graph by drawing the lines a bit thinner and showing the black line for wild type in the foreground. We prefer this version because the clustering of certain genotypes is clearly visible, which gets lost in the bar chart. We have included this version of Fig. 5b in the revised manuscript but are happy to replace it with the version below if that should be preferred.

Fig. 5 The DELLA proteins GAI and RGA are required for the CK-dependent regulation of vegetative phase change. **a** Number of leaves without abaxial trichomes of SD-grown *ahk2,3 gai rga*, *ahk2,3 toe1,2*, and *ahk2,3 gai rga toe1,2* hybrid plants in comparison to their respective parents and wild type. In box plots, the center line represents the median value and the boundaries indicate the 25th percentile (upper) and the 75th percentile (lower). The X marks the mean value. Whiskers extend to the largest and smallest value, excluding outliers which are shown as dots. **b** Length-to-width ratios of the blades of leaves 4 to 7. Data displayed are expressed as mean \pm SEM of SD-grown plants. Dots indicate each single biological replicate. Numbers of biological replicates: Col-0 (n = 36-38), *ahk2 ahk3* (n = 50-52), *gai rga* (n = 30-31), *ahk2,3 gai rga* (n = 41-42), *toe1 toe2* (n = 28-35), *ahk2,3 toe1,2* (n = 38-39), *gai rga toe1,2* (n = 27-29), *ahk2,3 gai rga toe1,2* (n = 49-53). Letters in **a** indicate statistically significant differences between the genotypes, as calculated by Kruskal-Wallis test ($q < 0.05$). Letters in **b** indicate statistically significant differences between the genotypes regarding individual leaves indicated by the subscript numbers, as calculated by one-way ANOVA, post-hoc Tukey's test ($p < 0.05$).

References

1. Werner, S., Bartrina, I. & Schmülling, T. Cytokinin regulates vegetative phase change in *Arabidopsis thaliana* through the miR172/TOE1-TOE2 module. *Nat. Commun.* **12**, 5816 (2021).
2. Aukerman, M. J. & Sakai, H. Regulation of flowering time and floral organ identity by a microRNA and its *APETALA2*-like target genes. *Plant Cell* **15**, 2730-2741 (2003).
3. Chen, X. M. A microRNA as a translational repressor of *APETALA2* in *Arabidopsis* flower development. *Science* **303**, 2022-2025 (2004).
4. Lauter, N., Kampani, A., Carlson, S., Goebel, M. & Moose, S. P. microRNA172 down-regulates *glossy15* to promote vegetative phase change in maize. *PNAS* **102**, 9412-9417 (2005).